# Mutual Information Estimation via Normalizing Flows

**Butakov I. D.**
Skoltech,[*] MIPT,[†] Sirius[‡]
butakov.id@phystech.su

**Tolmachev A. D.**
Skoltech, MIPT
tolmachev.ad@phystech.su

**Malanchuk S. V.**
MIPT, Skoltech
malanchuk.sv@phystech.su

**Neopryatnaya A. M.**
MIPT, Skoltech
neopryatnaya.am@phystech.su

**Frolov A. A.**
Skoltech
al.frolov@skoltech.ru

## Abstract

We propose a novel approach to the problem of *mutual information* (MI) estimation via introducing a family of estimators based on normalizing flows. The estimator maps original data to the target distribution, for which MI is easier to estimate. We additionally explore the target distributions with known closed-form expressions for MI. Theoretical guarantees are provided to demonstrate that our approach yields MI estimates for the original data. Experiments with high-dimensional data are conducted to highlight the practical advantages of the proposed method.

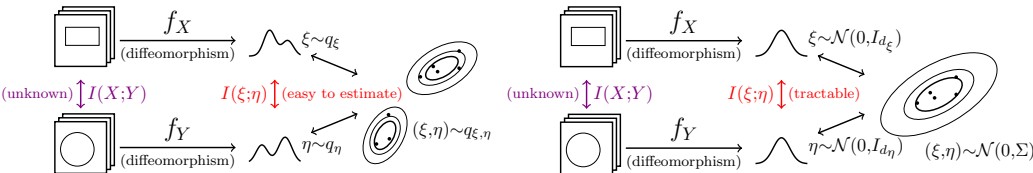

(a) MIENF (general base distribution).      (b) $\mathcal{N}$-MIENF (Gaussian base distribution).

Figure 1: We propose transforming a pair of random vectors (RVs) via a Cartesian product of learnable diffeomorphisms to facilitate mutual information (MI) estimation. Ideally, we achieve tractable MI in the latent space. As diffeomorphisms preserve information, MI between latent representations equals MI between the original RVs.

## 1 Introduction

Information-theoretic analysis of deep neural networks (DNN) has attracted recent interest due to intriguing fundamental results and new hypotheses. Applying information theory to DNNs may provide novel tools for explainable AI via estimation of information flows [1–5], as well as new ways to encourage models to extract and generalize information [1, 6–8]. Useful applications of information theory to the classical problem of independence testing are also worth noting [9–11].

---

[*]Skolkovo Institute of Science and Technology

[†]Moscow Institute of Physics and Technology

[‡]Sirius University of Science and Technology

38th Conference on Neural Information Processing Systems (NeurIPS 2024).

Most of the information theory applications to the field of machine learning are based on the two central information-theoretic quantities: *differential entropy* and *mutual information* (MI). The latter quantity is widely used as an invariant measure of the non-linear dependence between random variables, while differential entropy is usually viewed as a measure of randomness. However, as it has been shown in the previous works [12, 13], MI and differential entropy are extremely hard to estimate in the case of high-dimensional data. It is argued that such estimation is also challenging for long-tailed distributions and large values of MI [14]. These problems considerably limit the applications of information theory to real-scale machine learning problems. However, recent advances in the neural estimation methods show that complex parametric estimators achieve relative practical success in the cases where classical MI estimation techniques fail [7, 15–20].

This paper addresses the mentioned problem of the mutual information estimation in high dimensions via using normalizing flows [21–24]. Some recent works also utilize generative models to estimate MI. According to a general generative approach described in [16], generative models can be used to reconstruct probability density functions (PDFs) of marginal and joint distributions to estimate differential entropy and MI via a Monte Carlo (MC) integration. However, as it is mentioned in the original work, when flow-based generative models are used, the resulting estimates are poor even when the data is of simple structure. This approach is further investigated in [25]. The estimator proposed in [20] uses score-based diffusion models to estimate the differential entropy and MI without an explicit reconstruction of the PDFs. Increased accuracy of this estimator comes at a cost of training score networks and using them to compute an MC estimate of Kullback–Leibler divergence (KLD). Finally, in [11] normalizing flows are used to transform marginal distributions into Gaussian distributions, after which a zero-correlation criterion is employed to test the zero-MI hypothesis. The same idea is later used in [25] (see DINE-Gaussian) to acquire an MI estimate, but no corresponding error bounds are possible to derive, as knowing marginal distributions only is insufficient to calculate the MI (see Remark 4.5), which makes this estimator substantially flawed. We also note the work, where normalizing flows are combined with a $k$-NN entropy estimator [18].

In contrast, our method allows for simplified (cheap and low-variance MC integration is required) or even *direct* (i.e., no MC integration, nearest neighbors search or other similar data manipulations are required) and the accurate MI estimation with asymptotic and non-asymptotic error bounds. Our contributions in this work are the following:

1. We propose a MI-preserving technique to simplify the joint distribution of two random vectors (RVs) via a Cartesian product of trainable normalizing flows in order to facilitate the MI estimation. Non-asymptotic error bounds are provided, with the gap approaching zero under certain commonly satisfied assumptions, showing that our estimator is consistent.

2. We suggest restricting the proposed MI estimator to allow for a *direct* MI calculation via a *simple closed-form formula*. We further refine our approach to require only $O(d)$ additional learnable parameters to estimate the MI (here $d$ denotes the dimension of the data). We provide additional theoretical and statistical guarantees for our restricted estimator: variance and non-asymptotic error bounds are derived.

3. We validate and evaluate our method via experiments with high-dimensional synthetic data with known ground truth MI. We show that the proposed MI estimator performs well in comparison to the ground truth and some other advanced estimators during the tests with high-dimensional compressible and incompressible data of various complexity.

This article is organized as follows. In Section 2, the necessary background is provided and the key concepts of information theory are introduced. Section 3 describes the general method and corresponding theoretical results. In Section 4 we restrict our method to allow for accurate MI estimation via a closed-form formula. In Section 5, a series of experiments is performed to evaluate the proposed method and compare it to several other key MI estimators. Finally, the results are discussed in Section 6.

We provide all the proofs in Appendix A, additional details on the benchmarks we use in Appendix B, overfitting analysis in Appendix C, supplementary results regarding the information-based disentanglement of real data in Appendix D, and technical details in Appendix E.

## 2 Preliminaries

Consider random vectors, denoted as $X\colon \Omega \to \mathbb{R}^n$ and $Y\colon \Omega \to \mathbb{R}^m$, where $\Omega$ represents the sample space. Let us assume that these random vectors are absolutely continuous, having probability density functions (PDF) denoted as $p(x)$, $p(y)$, and $p(x,y)$, respectively, where the latter refers to the joint PDF. The differential entropy of $X$ is defined as follows:

$$h(X) = -\mathbb{E}\log p(x) = -\int_{\text{supp}\,X} p(x)\log p(x)\,dx,$$

where $\text{supp}\,X \subseteq \mathbb{R}^n$ represents the *support* of $X$, and $\log(\cdot)$ denotes the natural logarithm. Similarly, we define the joint differential entropy as $h(X,Y) = -\mathbb{E}\log p(x,y)$ and conditional differential entropy as $h(X \mid Y) = -\mathbb{E}\log p(X|Y) = -\mathbb{E}_Y\left(\mathbb{E}_{X|Y=y}\log p(X \mid Y=y)\right)$. Finally, the mutual information (MI) is given by $I(X;Y) = h(X) - h(X \mid Y)$, and the following equivalences hold

$$I(X;Y) = h(X) - h(X \mid Y) = h(Y) - h(Y \mid X), \tag{1}$$

$$I(X;Y) = h(X) + h(Y) - h(X,Y), \tag{2}$$

$$I(X;Y) = \mathrm{D}_{\mathrm{KL}}\left(p_{X,Y} \,\|\, p_X \otimes p_Y\right) \tag{3}$$

Mutual information can also be defined as an expectation of the *pointwise mutual information*:

$$\mathrm{PMI}_{X,Y}(x,y) = \log\left[\frac{p(x \mid y)}{p(x)}\right], \quad I(X;Y) = \mathbb{E}\,\mathrm{PMI}_{X,Y}(X,Y) \tag{4}$$

The above definitions can be generalized via Radon-Nikodym derivatives and induced densities in case of distributions supports being manifolds, see [26].

The differential entropy estimation is a separate classical statistical problem. Recent works have proposed several novel ways to acquire the estimate in the high-dimensional case [12, 18, 20, 27–30]. Due to Equation (2), mutual information can be found by estimating entropy values separately. In contrast, this paper suggests an approach that estimates MI values directly.

We also have to mention the well-known fundamental property of MI, which is invariance under smooth injective mappings. The following theorem appears in literature in slightly different forms [14, 19, 31–33]; we utilize the one, which is the most convenient to use with normalizing flows.

**Theorem 2.1.** *Let $\xi\colon \Omega \to \mathbb{R}^{n'}$ be an absolutely continuous random vector, and let $g\colon \mathbb{R}^{n'} \to \mathbb{R}^n$ be an injective piecewise-smooth mapping with Jacobian $J$, satisfying $n \geq n'$ and $\det\left(J^T J\right) \neq 0$ almost everywhere. Let PDFs $p_\xi$ and $p_{\xi|\eta}$ exist. Then*

$$\mathrm{PMI}_{\xi,\eta}(\xi,\eta) \overset{\text{a.s.}}{=} \mathrm{PMI}_{g(\xi),\eta}(g(\xi),\eta), \quad I(\xi;\eta) = I\left(g(\xi);\eta\right) \tag{5}$$

In our work, we heavily rely on the *normalizing flows* [23, 24] – trainable smooth bijective mappings with tractable Jacobian. However, to understand our results, it is sufficient to know that flow models (a) satisfy the conditions on $g$ in Theorem 2.1 *by definition*, (b) can model any absolutely continuous Borel probability measure (*universality* property) and (c) are trained via a likelihood maximization, which is equivalent to a Kullback-Leibler divergence minimization. For more details, we refer the reader to a more complete and rigorous overview of normalizing flows provided in [34].

## 3 General method

Our task is to estimate $I(X;Y)$, where $X, Y$ are random vectors. Here we focus on the absolutely continuous $(X,Y)$, as it is the most common case in practice. Note that Theorem 2.1 allows us to train normalizing flows $f_X, f_Y$, apply them to $X, Y$ and consider estimating MI between the latent representations, as $I(f_X(X); f_Y(Y)) = I(X;Y)$.

The key idea of our method is to train $f_X$ and $f_Y$ in such a way that $I(f_X(X); f_Y(Y))$ is easy to estimate. For example, one can hope to acquire tractable pointwise mutual information (PMI), which can be then averaged via MC integration [32]. Unfortunately, the PMI invariance (Theorem 2.1) restricts the possible distributions of $(f_X(X), f_Y(Y))$ to an unknown family, making the exact MI recovery via such technique unfeasible.

However, one can always approximate the PDF in latent space via a (preferably, simple) model $q \in \mathcal{Q}$ with tractable PMI, and train $q$, $f_X$ and $f_Y$ to minimize the discrepancy between the real and the proposed PMI. The complexity of $q$ serves as a tradeoff: by selecting a poor $\mathcal{Q}$, one might experience a considerable bias of the estimate; on the other hand, choosing $\mathcal{Q}$ to be a universal PDF approximation family, one acquires a consistent, but computationally expensive MI estimate. Flows $f_X$, $f_Y$ are used to tighten the approximation bound. We formalize this intuition in the following theorems:

**Theorem 3.1.** *Let $(\xi, \eta)$ be absolutely continuous with PDF $p_{\xi,\eta}$. Let $q_{\xi,\eta}$ be a PDF defined on the same space as $p_{\xi,\eta}$. Let $p_\xi$, $p_\eta$, $q_\xi$ and $q_\eta$ be the corresponding marginal PDFs. Then*

$$I(\xi;\eta) = \underbrace{\mathbb{E}_{\mathbb{P}_{\xi,\eta}} \log \left[ \frac{q_{\xi,\eta}(\xi,\eta)}{q_\xi(\xi)q_\eta(\eta)} \right]}_{I_q(\xi;\eta)} + D_{KL}\left(p_{\xi,\eta} \,\|\, q_{\xi,\eta}\right) - D_{KL}\left(p_\xi \otimes p_\eta \,\|\, q_\xi \otimes q_\eta\right) \quad (6)$$

**Corollary 3.2.** *Under the assumptions of Theorem 3.1, $|I(\xi;\eta) - I_q(\xi;\eta)| \leq D_{KL}\left(p_{\xi,\eta} \,\|\, q_{\xi,\eta}\right)$.*

This allows us to define the following MI estimate:

$$\hat{I}_{MIENF}(\{(x_k,y_k)\}_{k=1}^N) \triangleq \hat{I}_{\hat{q}}(\hat{f}_X(X); \hat{f}_Y(Y)) = \frac{1}{N}\sum_{k=1}^N \log\left[\frac{\hat{q}_{\xi,\eta}(\hat{f}_X(x_k), \hat{f}_Y(y_k))}{\hat{q}_\xi(\hat{f}_X(x_k))\hat{q}_\eta(\hat{f}_Y(y_k))}\right], \quad (7)$$

where $\{(x_k, y_k)\}_{k=1}^N$ is a sampling from $(X, Y)$, and $\hat{q}$, $\hat{f}_X$ and $\hat{f}_Y$ are selected according to the maximum likelihood. The latter makes $\hat{I}_{MIENF}$ a consistent estimator:

**Corollary 3.3** ($\hat{I}_{MIENF}$ is consistent). *Let $X$, $Y$, $\hat{f}_X^{-1}$ and $\hat{f}_Y^{-1}$ satisfy the conditions of Theorem 2.1. Let $\{(x_k, y_k)\}_{k=1}^N$ be an i.i.d. sampling from $(X, Y)$. Let $\mathcal{Q}$ be a family of universal PDF approximators for a class of densities containing $\mathbb{P}_{X,Y} \circ (f_X^{-1} \times f_Y^{-1})$ (pushforward probability measure in the latent space), meaning the convergence in probability of a maximum-likelihood estimate from $\mathcal{Q}$ to the ground-truth distribution if $N$ increases. Let $\hat{q}_N \in \mathcal{Q}$ be a maximum-likelihood estimate of $\mathbb{P}_{X,Y} \circ (f_X^{-1} \times f_Y^{-1})$ from the samples $\{(f_X(x_k), f_Y(y_k))\}_{k=1}^N$. Let $I_{\hat{q}_N}(f_X(X); f_Y(Y))$ exist for every $N$. Then*

$$\hat{I}_{MIENF}(\{(x_k, y_k)\}_{k=1}^N) \xrightarrow[N\to\infty]{\mathbb{P}} I(X;Y)$$

Note that maximum-likelihood training of $f_X$, $f_Y$ also minimizes $D_{KL}\left(\mathbb{P}_{X,Y} \circ (f_X^{-1} \times f_Y^{-1}) \,\|\, \hat{q}_{\xi,\eta}\right)$, which allows for surprisingly simple $q \in \mathcal{Q}$ to be used, as we show in the subsequent sections.

The described approach is as general as possible. We use it as a starting point for a development of a more elegant, cheap and practically sound MI estimator. We also do not incorporate conditions, under which the universality property of $\mathcal{Q}$ holds, as they depend on the choice of $\mathcal{Q}$; if one is interested in using normalizing flows as $\mathcal{Q}$, we refer to Section 3.4.3 in [34] or to [25] for more details.

## 4 Using Gaussian base distribution

Note that the general approach requires finding the maximum-likelihood estimate $\hat{q}$ and using it to perform an MC integration to acquire $\hat{I}_{\hat{q}}(f_X(X); f_Y(Y))$.

In this section, we drop these requirements by restricting our estimator via choosing $\mathcal{Q}$ to be a family of multivariate Gaussian PDFs. This allows us (a) to *directly* estimate the MI via a *closed-form* expression, (b) to employ a closed-form expression for optimal $\hat{q}$, (c) to leverage the maximum entropy principle for Gaussian distributions, thus acquiring better non-asymptotic bounds, and (d) to analyze the variance of the proposed estimate.

**Theorem 4.1** (Theorem 8.6.5 in [35]). *Let $Z$ be a $d$-dimensional absolutely continuous random vector with probability density function $p_Z$, mean $m$ and covariance matrix $\Sigma$. Then*

$$h(Z) = h\left(\mathcal{N}(m, \Sigma)\right) - D_{KL}\left(p_Z \,\|\, \mathcal{N}(m, \Sigma)\right) = \frac{d}{2}\log(2\pi e) + \frac{1}{2}\log \det \Sigma - D_{KL}\left(p_Z \,\|\, \mathcal{N}(m, \Sigma)\right)$$

*Remark* 4.2. Note that $h(Z)$ may not be equal to $h(Z') - D_{KL}\left(p_Z \,\|\, p_{Z'}\right)$ for arbitrary $Z'$.

**Corollary 4.3.** *Let $(\xi, \eta)$ be an absolutely continuous pair of random vectors with joint and marginal probability density functions $p_{\xi,\eta}$, $p_\xi$ and $p_\eta$ correspondingly, and mean and covariance matrix being*

$$m = \begin{bmatrix} m_\xi \\ m_\eta \end{bmatrix}, \quad \Sigma = \begin{bmatrix} \Sigma_{\xi,\xi} & \Sigma_{\xi,\eta} \\ \Sigma_{\eta,\xi} & \Sigma_{\eta,\eta} \end{bmatrix}$$

*Then*

$$I(\xi; \eta) = \frac{1}{2} \left[ \log \det \Sigma_{\xi,\xi} + \log \det \Sigma_{\eta,\eta} - \log \det \Sigma \right] + $$
$$+ \, \mathrm{D}_{\mathrm{KL}} \left( p_{\xi,\eta} \, \| \, \mathcal{N}(m, \Sigma) \right) - \mathrm{D}_{\mathrm{KL}} \left( p_\xi \otimes p_\eta \, \| \, \mathcal{N}(m, \mathrm{diag}(\Sigma_{\xi,\xi}, \Sigma_{\eta,\eta})) \right),$$

*which implies the following in the case of marginally Gaussian $\xi$ and $\eta$:*

$$I(\xi; \eta) \geq \frac{1}{2} \left[ \log \det \Sigma_{\xi,\xi} + \log \det \Sigma_{\eta,\eta} - \log \det \Sigma \right], \tag{8}$$

*with the equality holding if and only if $(\xi, \eta)$ are jointly Gaussian.*

**Corollary 4.4.** *Under the assumptions of Corollary 4.3,*

$$\left| I(\xi; \eta) - \frac{1}{2} \left[ \log \det \Sigma_{\xi,\xi} + \log \det \Sigma_{\eta,\eta} - \log \det \Sigma \right] \right| \leq \mathrm{D}_{\mathrm{KL}} \left( p_{\xi,\eta} \, \| \, \mathcal{N}(m, \Sigma) \right).$$

*Remark 4.5.* The upper bound from Corollary 4.4 is tight, consider $\xi \sim \mathcal{N}(0, 1)$, $\eta = (2B - 1) \cdot \xi$, where $B \sim \mathrm{Bernoulli}(1/2)$ and is independent of $\xi$.

From now on we denote $f_X(X)$ and $f_Y(Y)$ as $\xi$ and $\eta$ correspondingly. Note that, in contrast to Theorem 3.1 and Corollary 3.2, $I_q(\xi; \eta)$ is replaced by a closed-form expression, which is not possible to achieve in general. The provided closed-form expression allows for calculating MI for jointly Gaussian $(\xi, \eta)$, and serves as a lower bound on MI in the general case of $\xi$ and $\eta$ being only marginally Gaussian.

## 4.1 General binormalization approach

In order to minimize $\mathrm{D}_{\mathrm{KL}} \left( p_{\xi,\eta} \, \| \, \mathcal{N}(m, \Sigma) \right)$, we train $f_X \times f_Y$ as a single normalizing flow. Instead of maximizing the log-likelihood using two separate and fixed base (latent) distributions, we maximize the log-likelihood of the joint sampling $\{(x_k, y_k)\}_{k=1}^N$ using the whole set of Gaussian distributions as possible base distributions.

**Definition 4.6.** We denote a set of $d$-dimensional Gaussian distributions as $S_{\mathcal{N}}^d \triangleq \left\{ \mathcal{N}(m, \Sigma) \mid m \in \mathbb{R}^d, \Sigma \in \mathbb{R}^{d \times d} \right\}$.[4]

**Definition 4.7.** The log-likelihood of a sampling $\{z_k\}_{k=1}^N$ with respect to a set of absolutely continuous probability distributions $S$ is defined as follows:

$$\mathcal{L}_S(\{z_k\}) \triangleq \sup_{\mu \in S} \mathcal{L}_\mu(\{z_k\}) = \sup_{\mu \in S} \sum_{k=1}^N \log \left[ \left( \frac{d\mu}{dz} \right)(z_k) \right]$$

Let us define $f \triangleq f_X \times f_Y$ (Cartesian product of flows) and $S \circ f = \{\mu \circ f \mid \mu \in S\}$ (set of pushforward measures). In our case, $\mathcal{L}_{S_{\mathcal{N}} \circ f} \left( \{(x_k, y_k)\} \right)$ can be expressed in a closed-form via the change of variables formula (identically to a classical normalizing flows setup) and maximum-likelihood estimates for $m$ and $\Sigma$.

**Statement 4.8.**

$$\mathcal{L}_{S_{\mathcal{N}} \circ (f_X \times f_Y)} \left( \{(x_k, y_k)\} \right) = \log \left| \det \frac{\partial f(x, y)}{\partial(x, y)} \right| + \mathcal{L}_{\mathcal{N}(\hat{m}, \hat{\Sigma})} (\{f(x_k, y_k)\}),$$

*where*

$$\log \left| \det \frac{\partial f(x, y)}{\partial(x, y)} \right| = \log \left| \det \frac{\partial f_X(x)}{\partial x} \right| + \log \left| \det \frac{\partial f_Y(y)}{\partial y} \right|,$$

$$\hat{m} = \frac{1}{N} \sum_{k=1}^N f(x_k, y_k), \qquad \hat{\Sigma} = \frac{1}{N} \sum_{k=1}^N (f(x_k, y_k) - \hat{m})(f(x_k, y_k) - \hat{m})^T$$

---

[4]We omit $d$ whenever it can be deduced from the context.

Maximization of $\mathcal{L}_{S_{\mathcal{N}} \circ (f_X \times f_Y)}(\{(x_k, y_k)\})$ with respect to parameters of $f_X$ and $f_Y$ minimizes $D_{\mathrm{KL}}(p_{\xi,\eta} \| \mathcal{N}(m, \Sigma))$ [34], making it possible to apply Theorem 2.1 and Corollary 4.3 to acquire an MI estimate with corresponding non-asymptotic error bounds from Corollary 4.4:

$$\hat{I}_{\mathcal{N}\text{-MIENF}}(\{(x_k, y_k)\}_{k=1}^{N}) \triangleq \frac{1}{2} \left[ \log \det \hat{\Sigma}_{\xi,\xi} + \log \det \hat{\Sigma}_{\eta,\eta} - \log \det \hat{\Sigma} \right] \tag{9}$$

Note that if only marginal Gaussianization is achieved, Equation (9) serves as a lower bound estimate. As $\hat{I}_{\mathcal{N}\text{-MIENF}}$ involves maximum-likelihood estimates of covariance matrices, existing results can be employed to acquire the asymptotic variance:

**Lemma 4.9** (Lemma 2 in [25]). *Let $f_X$, $f_Y$ be fixed. Let $(\xi, \eta)$ have finite covariance matrix. Then, the asymptotic variance of $\hat{I}_{\mathcal{N}\text{-MIENF}}$ is $O(d^2/N)$, with $d$ being the dimensionality. If $(\xi, \eta)$ is also Gaussian, the asymptotic variance is further improved to $O(d/N)$.*

## 4.2 Refined approach

Although the proposed general method is compelling, as it requires only the slightest modifications to the conventional normalizing flow setup to make the application of the closed-form expressions for MI possible, we have to mention several drawbacks.

Firstly, the log-likelihood maximum is ambiguous, as $\mathcal{L}_{S_{\mathcal{N}}}$ is invariant under invertible affine mappings, which makes the proposed log-likelihood maximization an ill-posed problem:

*Remark* 4.10. Let $A \in \mathbb{R}^{d \times d}$ be a non-singular matrix, $b \in \mathbb{R}$, $\{z_k\}_{k=1}^{N} \subseteq \mathbb{R}^d$. Then

$$\mathcal{L}_{S_{\mathcal{N}} \circ (Az+b)}(\{z_k\}) = \mathcal{L}_{S_{\mathcal{N}}}(\{z_k\})$$

Secondly, this method requires a regular (ideally, after every gradient descent step) updating of $\hat{m}$ and $\hat{\Sigma}$ for the whole dataset, which is expensive. In practice, these estimates can be replaced with batchwise maximum likelihood estimates, which are used to update $\hat{m}$ and $\hat{\Sigma}$ via exponential moving average (EMA). This approach, however, requires tuning EMA multiplication coefficient in accordance with the learning rate to make the training fast yet stable. We also note that $\hat{m}$ and $\hat{\Sigma}$ can be made learnable via the gradient ascent, but the benefits of the closed-form expressions for $\mathcal{L}_{S_{\mathcal{N}}}$ in Statement 4.8 are thus lost.

Finally, each loss function evaluation requires inversion of $\hat{\Sigma}$, and each MI estimation requires evaluation of $\det \hat{\Sigma}$ and determinants of two diagonal blocks of $\hat{\Sigma}$. This might be resource-consuming in high-dimensional cases, as matrices may not be sparse. Numerical instabilities might also occur if $\hat{\Sigma}$ happens to be ill-conditioned (might happen in the case of data lying on a manifold or due to high MI).

That is why we propose an elegant and simple way to eliminate all the mentioned problems by further narrowing down $S_{\mathcal{N}}$ to a subclass of Gaussian distributions with simple and bounded covariance matrices and fixed means. This approach is somewhat reminiscent of the non-linear canonical correlation analysis [36, 37].

**Definition 4.11.**

$$S_{\text{tridiag-}\mathcal{N}}^{d_\xi, d_\eta} \triangleq \left\{ \mathcal{N}(0, \Sigma) \mid \Sigma_{\xi,\xi} = I_{d_\xi}, \Sigma_{\eta,\eta} = I_{d_\eta}, \Sigma_{\xi,\eta} (\equiv \Sigma_{\eta,\xi}^T) = \mathrm{diag}(\{\rho_j\})^{d_\xi \times d_\eta}, \rho_j \in [0; 1) \right\}$$

This approach solves all the aforementioned problems without any loss in generality, as it is shown by the following results:

**Corollary 4.12.** *If $(\xi, \eta) \sim \mu \in S_{\text{tridiag-}\mathcal{N}}$, then*

$$I(\xi; \eta) = -\frac{1}{2} \sum_j \log(1 - \rho_j^2) \tag{10}$$

**Statement 4.13** (Canonical correlation analysis). *Let $(\xi, \eta) \sim \mathcal{N}(m, \Sigma)$, where $\Sigma$ is non-singular. There exist invertible affine mappings $\varphi_\xi$, $\varphi_\eta$ such that $(\varphi_\xi(\xi), \varphi_\eta(\eta)) \sim \mu \in S_{\text{tridiag-}\mathcal{N}}$. Due to Theorem 2.1, the following also holds: $I(\xi; \eta) = I(\varphi_\xi(\xi); \varphi_\eta(\eta))$.*

**Statement 4.14.** *Let* $(\xi, \eta) \sim \mathcal{N}(0, \Sigma) \in S_{\text{tridiag-}\mathcal{N}}$, $\{z_k\}_{k=1}^N \subseteq \mathbb{R}^{d_\xi + d_\eta}$. *Then*

$$\mathcal{L}_{\mathcal{N}(0,\Sigma)}(\{z_k\}) = I(\xi; \eta) + \mathcal{L}_{\mathcal{N}(0,I)}(\{\Sigma^{-1/2} z_k\}),$$

*where (implying $\rho_j = 0$ for $j > \min\{d_\xi, d_\eta\}$)*

$$\Sigma^{-1/2} = \left[ \begin{array}{ccc|ccc} \alpha_j + \beta_j & & & \alpha_j - \beta_j & & \\ & \ddots & & & \ddots & \\ \hline \alpha_j - \beta_j & & & \alpha_j + \beta_j & & \\ & \ddots & & & \ddots & \end{array} \right] \quad \begin{array}{l} \alpha_j = \dfrac{1}{2\sqrt{1+\rho_j}} \\[2ex] \beta_j = \dfrac{1}{2\sqrt{1-\rho_j}} \end{array}$$

$$\underbrace{\hphantom{XXXXXXX}}_{d_\xi} \underbrace{\hphantom{XXXXXXX}}_{d_\eta}$$

*and $I(\xi; \eta)$ is calculated via* (10).

Maximization of $\mathcal{L}_{S_{\text{tridiag-}\mathcal{N}} \circ (f_X \times f_Y)}(\{(x_k, y_k)\})$ with respect to the parameters of $f_X$ and $f_Y$ yields the following MI estimate:

$$\hat{I}_{\text{tridiag-}\mathcal{N}\text{-MIENF}}(\{(x_k, y_k)\}_{k=1}^N) \triangleq -\frac{1}{2} \sum_j \log(1 - \hat{\rho}_j^2), \tag{11}$$

where $\hat{\rho}_j$ are the maximum-likelihood estimates of $\rho_j$.

### 4.3 Tractable error bounds

Note that Corollary 3.2 and Corollary 4.4 provide us with non-asymptotic, but untractable bounds. These bounds can be estimated via various KL divergence estimators [7, 20, 38–40]. However, this requires training an additional neural network, which is computationally expensive.

Conveniently, as the proposed method involves maximization of the likelihood, a cheap and tractable lower bound on the KL divergence can be obtained via an entropy-cross-entropy decomposition:

$$\mathrm{D}_{\mathrm{KL}}(p \,\|\, q) = \mathbb{E}_{Z \sim p} \log \frac{p(Z)}{q(Z)} = -\mathbb{E}_{Z \sim p} \log q(Z) - h(Z) \geq -\mathbb{E}_{Z \sim p} \log q(Z) - h(\mathcal{N}(m, \Sigma)) \tag{12}$$

Note that $\mathbb{E} \log q(Z)$ in Equation (12) is estimated by the log-likelihood of the samples in the latent space (which is inevitably evaluated each training step), and $h(Z)$ can be upper bounded by the entropy of a Gaussian distribution (see Theorem 4.1). Unfortunately, as Theorem 4.1 has already been employed to derive Corollary 4.3, the proposed bound is trivial (equaling to zero) in our Gaussian-based setup. However, this idea might still be useful in the general case.

### 4.4 Implementation details

In this section, we would like to emphasize the computational simplicity of the proposed amendments to the conventional normalizing flow setup.

Firstly, Statement 4.14 allows for a cheap log-likelihood computation and sample generation, as $\Sigma$, $\Sigma^{-1/2}$ and $\Sigma^{-1}$ are easily calculated from the $\{\rho_j\}$ and are sparse (tridiagonal, block-diagonalizable with $2 \times 2$ blocks). Secondly, the method requires only $d' = \min\{d_\xi, d_\eta\}$ additional parameters: the estimates for $\{\rho_j\}$. As $\rho_j \in [0; 1)$, an appropriate parametrization should be chosen to allow for stable learning of $\{\hat{\rho}_j\}$ via the gradient ascend. We propose using the *logarithm of cross-component MI*[5]: $w_j \triangleq \log I(\xi_j; \eta_j) = \log \left[ -\frac{1}{2} \log(1 - \rho_j^2) \right]$. In this parametrization $w_j \in \mathbb{R}$ and

$$\hat{I}_{\text{tridiag-}\mathcal{N}\text{-MIENF}}(\{(x_k, y_k)\}_{k=1}^N) = \sum_j e^{\hat{w}_j}, \quad \rho_j = \sqrt{1 - \exp(-2 \cdot e^{w_j})} \in (0; 1) \tag{13}$$

Although $\rho_j = 0$ is not achievable in the proposed parametrization, it can be made arbitrarily close to 0 with $w_j \to -\infty$.

---

[5]One can also consider the softplus parametrization, which allows to avoid the double exponentiation in (13). We, however, did not encounter any issues with the plain logarithm parametrization.

### 4.5 Extension to non-Gaussian base distributions and non-bijective flows

The proposed method can be easily generalized to account for any base distribution with closed-form expression for MI, or even a combination of such distributions. For example, a smoothed uniform distribution can be considered, with the learnable parameter being the smoothing constant $\varepsilon$, see Appendix B.2, Equation (14). However, due to Remark 4.2, neither Corollary 3.2, nor Corollary 4.4 can be used to bound the estimation error in this case.

Also note that, as Theorem 2.1 does not require $g$ to be bijective, our method is naturally extended to injective normalizing flows [41, 42]. Moreover, according to [19], such approach may actually facilitate the estimation of MI.

## 5 Experiments

To evaluate our estimator, we utilize synthetic datasets with known mutual information. In [14] and [19], extensive frameworks for evaluation of MI estimators have been proposed. In our work, we borrow complex high-dimensional tests from [19] and all non-Gaussian base distributions with known MI from [14] (see Appendix B for more details). The formal description of the dataset generation and estimator evaluation is provided in Algorithm 1. Essentially similar setups are widely used to test the MI estimators [7, 13, 19, 20, 31, 43].

---

**Algorithm 1** MI estimator evaluation

1: Generate two datasets of samples from random vectors $\xi$ and $\eta$ with known ground truth mutual information (see Corollary 4.3, Corollary 4.12 and Appendix B for examples): $\{(a_k, b_k)\}_{k=1}^{N}$.
2: Choose functions $g_\xi$ and $g_\eta$ satisfying conditions of Theorem 2.1, so $I(\xi; \eta) = I\big(g_\xi(\xi); g_\eta(\eta)\big)$.
3: Estimate $I\big(g_\xi(\xi); g_\eta(\eta)\big)$ via $\{(g_\xi(a_k), g_\eta(b_k))\}_{k=1}^{N}$; compare the result with the ground truth.

---

For the first set of experiments, we map a low-dimensional correlated Gaussian distribution to a distribution of high-dimensional images of geometric shapes (see Figure 2). We compare our method with the Mutual Information Neural Estimator (MINE) [7], Nguyen-Wainwright-Jordan (NWJ) [7, 39] and Nishiyama's [40] estimators, nearest neighbor Kraskov-Stoegbauer-Grassberger [31] and 5-nearest neighbors weighted Kozachenko-Leonenko (WKL) estimator [27, 44]; the latter is fed with the data compressed via a convolutional autoencoder (CNN AE) in accordance to the pipeline from [19].

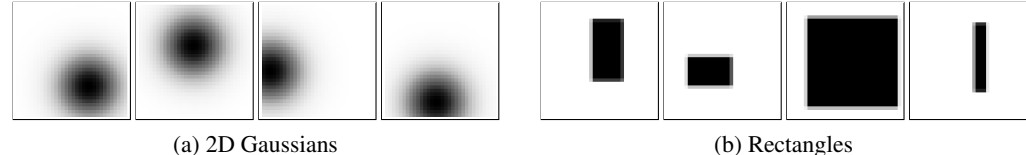

(a) 2D Gaussians        (b) Rectangles

Figure 2: Examples of synthetic images used in the tests. Note that images are high-dimensional, but admit latent structure, which is similar to real datasets.

For the second set of experiments, incompressible, high-dimensional non-Gaussian-based distributions are considered. These experiments are conducted to evaluate the robustness of our estimator to the distributions, which can not be precisely Gaussianized via a Cartesian product of two flows. In this section, we compare our method to the ground truth value only. We also provide a comparison with MINE and DINE-Gaussian [25] in Appendix B. For a more elaborate benchmarking of other estimators on these distributions, we refer the reader to [14].

For the tests with synthetic images, we use GLOW [45] normalizing flow architecture with $\log_2(\text{image size})$ splits, 2 blocks between splits and 16 hidden channels in each block, appended with a learnable orthogonal linear layer and a small $4$-layer Real NVP flow [46]. For the other tests, we use 6-layer Real NVP architecture. For further details (including the architecture of MINE critic network and CNN autoencoder), we refer the reader to Appendix E.

The results of the experiments performed with the high-dimensional synthetic images and non-Gaussian-based distribution are provided in Figure 3 and Figure 4 correspondingly.

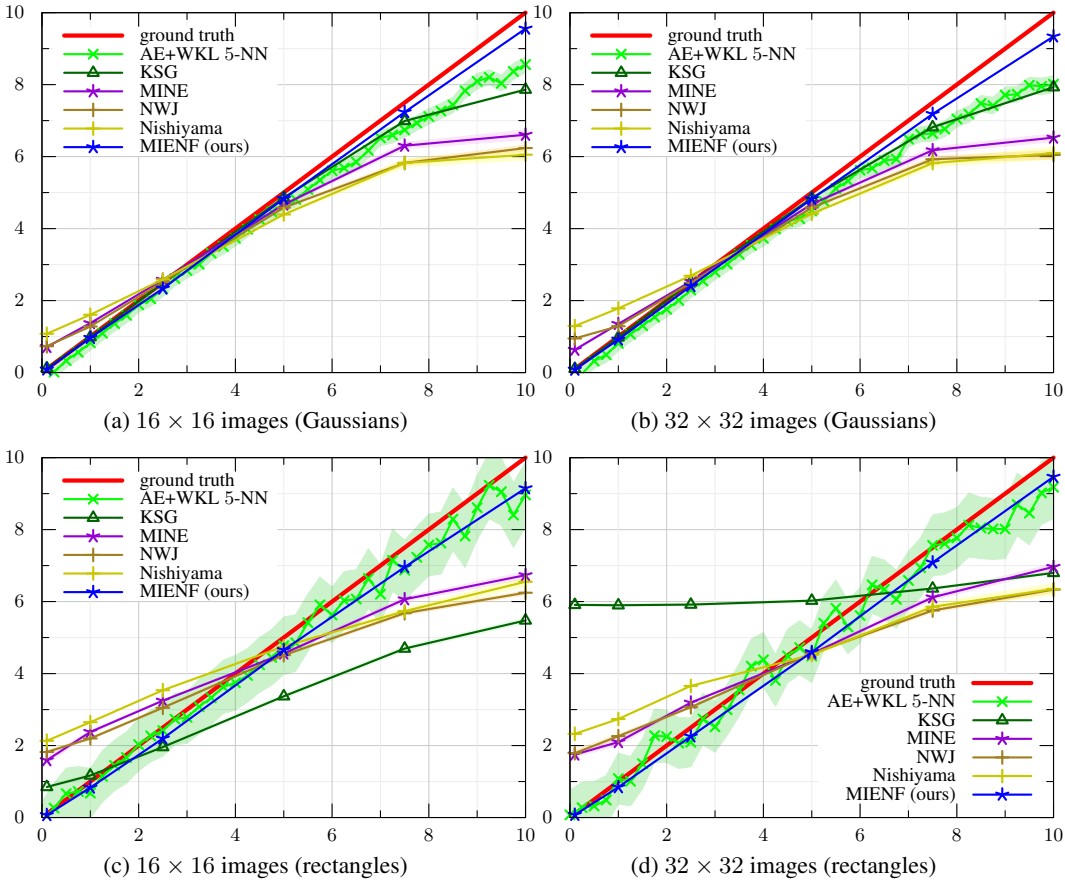

Figure 3: Comparison of the selected estimators. Along $x$ axes is $I(X;Y)$, along $y$ axes is $\hat{I}(X;Y)$. We plot 99.9% asymptotic CIs acquired either from the MC integration standard deviation (WKL, KSG) or from the epochwise averaging (other methods, 200 last epochs). $10 \cdot 10^3$ samples were used.

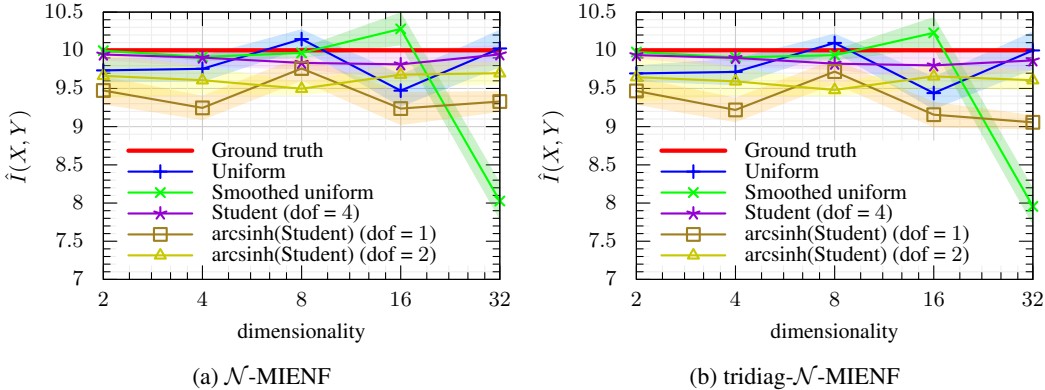

Figure 4: Tests with incompressible multidimensional data. "Uniform" denotes the uniformly distributed samples acquired from the correlated Gaussians via the Gaussian CDF. "Smoothed uniform" and "Student" denote the non-Gaussian-based distributions described in Appendix B. "arcsinh(Student)" denotes the arcsinh function applied to the "Student" example (this is done to avoid numerical instabilities in the case of long-tailed distributions). We run each test 5 times and plot 99.9% asymptotic Gaussian CIs. $10 \cdot 10^3$ samples were used. Note that $\mathcal{N}$-MIENF and tridiag-$\mathcal{N}$-MIENF yield almost the same results with similar bias.

We attribute the good performance of AE+WKL to the fact that the proposed synthetic datasets are easily and almost losslessly compressed via a CNN AE. We run additional experiments with much simpler, but incompressible data to show that the estimation error of WKL rapidly increases with the dimensionality. The results are provided in Table 1. In contrast, our method yields reasonable estimates in the same or similar cases presented in Figure 4.

Table 1: Evaluation of 5-NN weighted Kozachenko-Leonenko estimator on multidimensional uniformly distributed data. For each dimension $d_X = d_Y$, 11 estimates of MI are acquired with the ground truth ranging from 0 to 10 with a fixed step. The RMSE of the estimated MI relative to the ground-truth MI is calculated for each set of estimates.

| $d_{X,Y}$ | 2 | 4 | 8 | 16 | 32 | 64 |
|---|---|---|---|---|---|---|
| RMSE | 2.2 | 1.0 | 127.9 | 227.5 | 522.4 | 336.2 |

Overall, the proposed estimator performs well during all the experiments, including the incompressible high-dimensional data, large MI values and non-Gaussian-based tests. In Appendix D, we also apply our method to acquire disentengled representations of real data. Additionally, we give a brief commentary on the sample complexity of the proposed method and other NN-based estimators in Appendix C.

# 6   Discussion

Information-theoretic analysis of deep neural networks is a novel and developing approach. As it relies on a well-established theory of information, it potentially can provide fundamental, robust and intuitive results [47, 48]. Currently, this analysis is complicated due to main information-theoretic qualities — *differential entropy* and *mutual information* — being hard to measure in the case of high-dimensional data.

We have shown that it is possible to modify the conventional normalizing flow setup to harness all the benefits of simple and robust closed-form expressions for mutual information. Non-asymptotic error bounds for both variants of our method are derived, asymptotic variance and consistency analysis is carried out. We provide useful theoretical and practical insights on using the proposed method effectively. We have demonstrated the effectiveness of our estimator in various settings, including compressible and incompressible high-dimensional data, high values of mutual information and the data not acquired from the Gaussian distribution via invertible mappings.

Finally, it is worth noting that despite normalizing flows and Gaussian base distributions being used throughout our work, the proposed method can be extended to any type of base distribution with closed-form expression for mutual information and to any injective generative model. For example, a subclass of diffusion models can be considered [49, 50]. Injective normalizing flows [41, 42] are also compatible with the proposed pipeline. Gaussian mixtures can also be used as base distributions due to a relatively cheap MI calculation and the universality property [32].

**Limitations**   The main limitation of the general method is the ambiguity of $\mathcal{Q}$ (the family of PDF estimators used to estimate MI in the latent space), which can be either rich (yielding a consistent, but possibly expensive estimator), or poor (leading to the inconsistency of the estimate). However, in [32] it is argued that mixture models can achieve rather good tradeoff between the quality and the cost of a PMI approximation.

The major limitation of $\mathcal{N}$-MIENF is that its consistency is proven only for a certain class of distributions: the probability distribution should be equivalent to a Gaussian via a Cartesian product of diffeomorphisms. However, mathematical simplicity, rigorous bounds, low variance and relative practical success of the estimator suggest that the proposed method achieves a decent tradeoff.

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

# A  Complete proofs

**Theorem 2.1.** *Let $\xi\colon \Omega \to \mathbb{R}^{n'}$ be an absolutely continuous random vector, and let $g\colon \mathbb{R}^{n'} \to \mathbb{R}^n$ be an injective piecewise-smooth mapping with Jacobian $J$, satisfying $n \geq n'$ and $\det\left(J^T J\right) \neq 0$ almost everywhere. Let PDFs $p_\xi$ and $p_{\xi|\eta}$ exist. Then*

$$\mathrm{PMI}_{\xi,\eta}(\xi,\eta) \overset{\text{a.s.}}{=} \mathrm{PMI}_{g(\xi),\eta}(g(\xi),\eta), \quad I(\xi;\eta) = I\left(g(\xi);\eta\right) \tag{5}$$

*Proof of Theorem 2.1.* For any function $g$, let us denote $\sqrt{\det\left(J^T(x)J(x)\right)}$ (area transformation coefficient) by $\alpha(x)$ where it exists.

Foremost, let us note that in both cases, $p_\xi(x \mid \eta)$ and $p_{g(\xi)}(x' \mid \eta) = p_\xi(x \mid \eta)/\alpha(x)$ exist. Hereinafter, we integrate over $\operatorname{supp}\xi \cap \{x \mid \alpha(x) \neq 0\}$ instead of $\operatorname{supp}\xi$; as $\alpha \neq 0$ almost everywhere by the assumption, the values of the integrals are not altered.

According to the definition of the differential entropy,

$$h(g(\xi)) = -\int \frac{p_\xi(x)}{\alpha(x)} \log\left(\frac{p_\xi(x)}{\alpha(x)}\right) \alpha(x)\, dx =$$

$$= -\int p_\xi(x) \log\left(p_\xi(x)\right) dx + \int p_\xi(x) \log\left(\alpha(x)\right) dx =$$

$$= h(\xi) + \mathbb{E} \log \alpha(\xi).$$

$$h(g(\xi) \mid \eta) = \mathbb{E}_\eta\left(-\int \frac{p_\xi(x \mid \eta)}{\alpha(x)} \log\left(\frac{p_\xi(x \mid \eta)}{\alpha(x)}\right) \alpha(x)\, dx\right) =$$

$$= \mathbb{E}_\eta\left(-\int p_\xi(x \mid \eta) \log\left(p_\xi(x \mid \eta)\right) dx + \int p_\xi(x \mid \eta) \log\left(\alpha(x)\right) dx\right) =$$

$$= h(\xi \mid \eta) + \mathbb{E} \log \alpha(\xi)$$

Finally, by the MI definition,

$$I(g(\xi);\eta) = h(g(\xi)) - h(g(\xi) \mid \eta) = h(\xi) - h(\xi \mid \eta) = I(\xi;\eta).$$

Dropping the expectations/integrals in the equations above yields the proof of the PMI invariance. $\square$

**Theorem 3.1.** *Let $(\xi,\eta)$ be absolutely continuous with PDF $p_{\xi,\eta}$. Let $q_{\xi,\eta}$ be a PDF defined on the same space as $p_{\xi,\eta}$. Let $p_\xi$, $p_\eta$, $q_\xi$ and $q_\eta$ be the corresponding marginal PDFs. Then*

$$I(\xi;\eta) = \underbrace{\mathbb{E}_{\mathbb{P}_{\xi,\eta}} \log\left[\frac{q_{\xi,\eta}(\xi,\eta)}{q_\xi(\xi)q_\eta(\eta)}\right]}_{I_q(\xi;\eta)} + \mathrm{D}_{\mathrm{KL}}\left(p_{\xi,\eta} \,\|\, q_{\xi,\eta}\right) - \mathrm{D}_{\mathrm{KL}}\left(p_\xi \otimes p_\eta \,\|\, q_\xi \otimes q_\eta\right) \tag{6}$$

*Proof of Theorem 3.1.* In the following text, all the expectations are in terms of $\mathbb{P}_{\xi,\eta}$.

$$I(\xi;\eta) = \mathbb{E} \log\left[\frac{p_{\xi,\eta}(\xi,\eta)}{p_\xi(\xi)p_\eta(\eta)}\right] = \mathbb{E} \log\left[\frac{q_{\xi,\eta}(\xi,\eta)}{q_\xi(\xi)q_\eta(\eta)} \cdot \frac{p_{\xi,\eta}(\xi,\eta)}{q_{\xi,\eta}(\xi,\eta)} \cdot \frac{q_\xi(\xi)q_\eta(\eta)}{p_\xi(\xi)p_\eta(\eta)}\right] =$$

$$= I_q(\xi;\eta) + \mathbb{E} \log\left[\frac{p_{\xi,\eta}(\xi,\eta)}{q_{\xi,\eta}(\xi,\eta)}\right] + \mathbb{E} \log\left[\frac{q_\xi(\xi)}{p_\xi(\xi)}\right] + \mathbb{E} \log\left[\frac{q_\eta(\eta)}{p_\eta(\eta)}\right] =$$

$$= I_q(\xi;\eta) + \mathrm{D}_{\mathrm{KL}}\left(p_{\xi,\eta} \,\|\, q_{\xi,\eta}\right) - \mathrm{D}_{\mathrm{KL}}\left(p_\xi \otimes p_\eta \,\|\, q_\xi \otimes q_\eta\right)$$

$\square$

**Corollary 3.2.** *Under the assumptions of Theorem 3.1, $|I(\xi;\eta) - I_q(\xi;\eta)| \leq \mathrm{D}_{\mathrm{KL}}\left(p_{\xi,\eta} \,\|\, q_{\xi,\eta}\right)$.*

*Proof of Corollary 3.2.* As $\mathrm{D}_{\mathrm{KL}}\left(p_\xi \otimes p_\eta \,\|\, q_\xi \otimes q_\eta\right) \geq 0$,

$$I(\xi;\eta) \leq I_q(\xi,\eta) + \mathrm{D}_{\mathrm{KL}}\left(p_{\xi,\eta} \,\|\, q_{\xi,\eta}\right)$$

As $D_{\mathrm{KL}}\left(p_{\xi,\eta}\,\|\,q_{\xi,\eta}\right) \geq D_{\mathrm{KL}}\left(p_\xi\,\|\,q_\xi\right)$ and $D_{\mathrm{KL}}\left(p_{\xi,\eta}\,\|\,q_{\xi,\eta}\right) \geq D_{\mathrm{KL}}\left(p_\eta\,\|\,q_\eta\right)$ (monotonicity property, see Theorem 2.16 in [33]),

$$I(\xi;\eta) \geq I_q(\xi;\eta) + D_{\mathrm{KL}}\left(p_{\xi,\eta}\,\|\,q_{\xi,\eta}\right) - 2 \cdot D_{\mathrm{KL}}\left(p_{\xi,\eta}\,\|\,q_{\xi,\eta}\right) = I_q(\xi;\eta) - D_{\mathrm{KL}}\left(p_{\xi,\eta}\,\|\,q_{\xi,\eta}\right)$$

$\square$

**Corollary 3.3** ($\hat{I}_{\mathrm{MIENF}}$ is consistent). *Let $X$, $Y$, $\hat{f}_X^{-1}$ and $\hat{f}_Y^{-1}$ satisfy the conditions of Theorem 2.1. Let $\{(x_k, y_k)\}_{k=1}^N$ be an i.i.d. sampling from $(X, Y)$. Let $\mathcal{Q}$ be a family of universal PDF approximators for a class of densities containing $\mathbb{P}_{X,Y} \circ (f_X^{-1} \times f_Y^{-1})$ (pushforward probability measure in the latent space), meaning the convergence in probability of a maximum-likelihood estimate from $\mathcal{Q}$ to the ground-truth distribution if $N$ increases. Let $\hat{q}_N \in \mathcal{Q}$ be a maximum-likelihood estimate of $\mathbb{P}_{X,Y} \circ (f_X^{-1} \times f_Y^{-1})$ from the samples $\{(f_X(x_k), f_Y(y_k))\}_{k=1}^N$. Let $I_{\hat{q}_N}(f_X(X); f_Y(Y))$ exist for every $N$. Then*

$$\hat{I}_{\mathrm{MIENF}}(\{(x_k, y_k)\}_{k=1}^N) \xrightarrow[N\to\infty]{\mathbb{P}} I(X;Y)$$

*Proof of Corollary 3.3.* Following the assumptions on $\hat{q}_N$, $D_{\mathrm{KL}}\left(\mathbb{P}_{X,Y} \circ (f_X^{-1} \times f_Y^{-1})\,\|\,(\hat{q}_N)_{\xi,\eta}\right) \xrightarrow[N\to\infty]{\mathbb{P}}$ 0 (*universality* property). Due to Corollary 3.2, this ensures $I_{\hat{q}_N}(f_X(X); f_Y(Y)) \xrightarrow[N\to\infty]{\mathbb{P}}$ $I(f_X(X); f_Y(Y)) = I(X;Y)$ (the latter equality is due to Theorem 2.1). Finally, $\hat{I}_{\mathrm{MIENF}}(X;Y) \xrightarrow[N\to\infty]{\mathbb{P}} I_{\hat{q}_N}(f_X(X); f_Y(Y))$ as an MC estimate. $\square$

**Theorem 4.1** (Theorem 8.6.5 in [35]). *Let $Z$ be a $d$-dimensional absolutely continuous random vector with probability density function $p_Z$, mean $m$ and covariance matrix $\Sigma$. Then*

$$h(Z) = h\left(\mathcal{N}(m, \Sigma)\right) - D_{\mathrm{KL}}\left(p_Z\,\|\,\mathcal{N}(m, \Sigma)\right) = \frac{d}{2}\log(2\pi e) + \frac{1}{2}\log\det\Sigma - D_{\mathrm{KL}}\left(p_Z\,\|\,\mathcal{N}(m, \Sigma)\right)$$

*Proof of Theorem 4.1.* As $h(Z - m) = h(Z)$, let us consider a centered random vector $Z$. We denote probability density function of $\mathcal{N}(0, \Sigma)$ as $\phi_\Sigma$.

$$D_{\mathrm{KL}}\left(p_Z\,\|\,\mathcal{N}(0, \Sigma)\right) = \int_{\mathbb{R}^d} p_Z(z) \log \frac{p_Z(z)}{\phi_\Sigma(z)}\, dz = -h(Z) - \int_{\mathbb{R}^d} p_Z(z) \log \phi_\Sigma(z)\, dz$$

Note that

$$\int_{\mathbb{R}^d} p_Z(z) \log \phi_\Sigma(z)\, dz = const + \frac{1}{2}\mathbb{E}_Z\, z^T \Sigma^{-1} z = const + \frac{1}{2}\mathbb{E}_{\mathcal{N}(0,\Sigma)}\, z^T \Sigma^{-1} z = \int_{\mathbb{R}^d} \phi_\Sigma(z) \log \phi_\Sigma(z)\, dz,$$

from which we arrive at the final result:

$$D_{\mathrm{KL}}\left(p_Z\,\|\,\mathcal{N}(0, \Sigma)\right) = -h(Z) + h(\mathcal{N}(0, \Sigma))$$

$\square$

**Corollary 4.3.** *Let $(\xi, \eta)$ be an absolutely continuous pair of random vectors with joint and marginal probability density functions $p_{\xi,\eta}$, $p_\xi$ and $p_\eta$ correspondingly, and mean and covariance matrix being*

$$m = \begin{bmatrix} m_\xi \\ m_\eta \end{bmatrix}, \quad \Sigma = \begin{bmatrix} \Sigma_{\xi,\xi} & \Sigma_{\xi,\eta} \\ \Sigma_{\eta,\xi} & \Sigma_{\eta,\eta} \end{bmatrix}$$

*Then*

$$I(\xi;\eta) = \frac{1}{2}\left[\log\det\Sigma_{\xi,\xi} + \log\det\Sigma_{\eta,\eta} - \log\det\Sigma\right] +$$
$$+ D_{\mathrm{KL}}\left(p_{\xi,\eta}\,\|\,\mathcal{N}(m, \Sigma)\right) - D_{\mathrm{KL}}\left(p_\xi \otimes p_\eta\,\|\,\mathcal{N}(m, \mathrm{diag}(\Sigma_{\xi,\xi}, \Sigma_{\eta,\eta}))\right),$$

*which implies the following in the case of marginally Gaussian $\xi$ and $\eta$:*

$$I(\xi;\eta) \geq \frac{1}{2}\left[\log\det\Sigma_{\xi,\xi} + \log\det\Sigma_{\eta,\eta} - \log\det\Sigma\right], \tag{8}$$

*with the equality holding if and only if $(\xi, \eta)$ are jointly Gaussian.*

*Proof of Corollary 4.3.* By applying Theorem 4.1 to Equation (2), we derive the following expression:

$$I(\xi; \eta) = \frac{1}{2} \left[\log \det \Sigma_{\xi,\xi} + \log \det \Sigma_{\eta,\eta} - \log \det \Sigma\right] +$$
$$+ \mathrm{D}_{\mathrm{KL}} \left(p_{\xi,\eta} \,\|\, \mathcal{N}(m, \Sigma)\right) - \mathrm{D}_{\mathrm{KL}} \left(p_\xi \,\|\, \mathcal{N}(m_\xi, \Sigma_{\xi,\xi})\right) - \mathrm{D}_{\mathrm{KL}} \left(p_\eta \,\|\, \mathcal{N}(m_\eta, \Sigma_{\eta,\eta})\right)$$

Note that

$$\mathrm{D}_{\mathrm{KL}} \left(p_\xi \,\|\, \mathcal{N}(m_\xi, \Sigma_{\xi,\xi})\right) + \mathrm{D}_{\mathrm{KL}} \left(p_\eta \,\|\, \mathcal{N}(m_\eta, \Sigma_{\eta,\eta})\right) = \mathrm{D}_{\mathrm{KL}} \left(p_\xi \otimes p_\eta \,\|\, \mathcal{N}(m, \mathrm{diag}(\Sigma_{\xi,\xi}, \Sigma_{\eta,\eta}))\right),$$

which results in

$$I(\xi; \eta) = \frac{1}{2} \left[\log \det \Sigma_{\xi,\xi} + \log \det \Sigma_{\eta,\eta} - \log \det \Sigma\right] +$$
$$+ \mathrm{D}_{\mathrm{KL}} \left(p_{\xi,\eta} \,\|\, \mathcal{N}(m, \Sigma)\right) - \mathrm{D}_{\mathrm{KL}} \left(p_\xi \otimes p_\eta \,\|\, \mathcal{N}(m, \mathrm{diag}(\Sigma_{\xi,\xi}, \Sigma_{\eta,\eta}))\right)$$

$\square$

**Corollary 4.4.** *Under the assumptions of Corollary 4.3,*

$$\left| I(\xi; \eta) - \frac{1}{2} \left[\log \det \Sigma_{\xi,\xi} + \log \det \Sigma_{\eta,\eta} - \log \det \Sigma\right] \right| \leq \mathrm{D}_{\mathrm{KL}} \left(p_{\xi,\eta} \,\|\, \mathcal{N}(m, \Sigma)\right).$$

*Proof of Corollary 4.4.* The same as for Corollary 3.2 $\square$

**Statement 4.8.**

$$\mathcal{L}_{S_{\mathcal{N}} \circ (f_X \times f_Y)} \left(\{(x_k, y_k)\}\right) = \log \left| \det \frac{\partial f(x, y)}{\partial (x, y)} \right| + \mathcal{L}_{\mathcal{N}(\hat{m}, \hat{\Sigma})}(\{f(x_k, y_k)\}),$$

*where*

$$\log \left| \det \frac{\partial f(x, y)}{\partial (x, y)} \right| = \log \left| \det \frac{\partial f_X(x)}{\partial x} \right| + \log \left| \det \frac{\partial f_Y(y)}{\partial y} \right|,$$

$$\hat{m} = \frac{1}{N} \sum_{k=1}^{N} f(x_k, y_k), \qquad \hat{\Sigma} = \frac{1}{N} \sum_{k=1}^{N} (f(x_k, y_k) - \hat{m})(f(x_k, y_k) - \hat{m})^T$$

*Proof of Statement 4.8.* Due to the change of variables formula,

$$\mathcal{L}_{S_{\mathcal{N}} \circ (f_X \times f_Y)} \left(\{(x_k, y_k)\}\right) = \log \left| \det \frac{\partial f(x, y)}{\partial (x, y)} \right| + \mathcal{L}_{\mathcal{N}}(\{f(x_k, y_k)\})$$

As $f = f_X \times f_Y$, the Jacobian matrix $\frac{\partial f(x,y)}{\partial (x,y)}$ is block-diagonal, so

$$\log \left| \det \frac{\partial f(x, y)}{\partial (x, y)} \right| = \log \left| \det \frac{\partial f_X(x)}{\partial x} \right| + \log \left| \det \frac{\partial f_Y(y)}{\partial y} \right|$$

Finally, we use the maximum-likelihood estimates for $m$ and $\Sigma$ to drop the supremum in $\mathcal{L}_{\mathcal{N}}(\{f(x_k, y_k)\})$:

$$\hat{m} = \frac{1}{N} \sum_{k=1}^{N} f(x_k, y_k), \quad \hat{\Sigma} = \frac{1}{N} \sum_{k=1}^{N} (f(x_k, y_k) - \hat{m})(f(x_k, y_k) - \hat{m})^T \implies$$
$$\implies \mathcal{L}_{\mathcal{N}}(\{f(x_k, y_k)\}) = \mathcal{L}_{\mathcal{N}(\hat{m}, \hat{\Sigma})}(\{f(x_k, y_k)\})$$

$\square$

**Lemma 4.9** (Lemma 2 in [25])**.** *Let $f_X$, $f_Y$ be fixed. Let $(\xi, \eta)$ have finite covariance matrix. Then, the asymptotic variance of $\hat{I}_{\mathcal{N}\text{-MIENF}}$ is $O(d^2/N)$, with $d$ being the dimensionality. If $(\xi, \eta)$ is also Gaussian, the asymptotic variance is further improved to $O(d/N)$.*

*Proof of Lemma 4.9.* The variance of $\hat{I}_{\mathcal{N}\text{-MIENF}}$ is upper bounded by the sum of the variances of the log-det terms. If $(\xi, \eta)$ is Gaussian, the log-det terms are asymptotically normal with the asymptotic variance being $2d/N$ (Corollary 1 in [51]). If $(\xi, \eta)$ is not Gaussian, the central limit theorem can be applied to each element of the covariance matrix, which in combination with the delta method yields the final result. $\square$

*Remark* 4.10. Let $A \in \mathbb{R}^{d \times d}$ be a non-singular matrix, $b \in \mathbb{R}$, $\{z_k\}_{k=1}^{N} \subseteq \mathbb{R}^d$. Then

$$\mathcal{L}_{S_{\mathcal{N}} \circ (Az+b)}(\{z_k\}) = \mathcal{L}_{S_{\mathcal{N}}}(\{z_k\})$$

*Proof of Remark 4.10.*

$$\mathcal{L}_{S_{\mathcal{N}} \circ (Az+b)}(\{z_k\}) = \log|\det A| + \mathcal{L}_{S_{\mathcal{N}}}(\{Az_k+b\}) = \log|\det A| + \mathcal{L}_{\mathcal{N}(A\hat{m}+b, A\hat{\Sigma}A^T)}(\{Az_k+b\}) =$$
$$= \log|\det A| + \log|\det A^{-1}| + \mathcal{L}_{\mathcal{N}(\hat{m}, \hat{\Sigma})}(\{z_k\}) = \mathcal{L}_{S_{\mathcal{N}}}(\{z_k\})$$

$\square$

**Corollary 4.12.** *If $(\xi, \eta) \sim \mu \in S_{\text{tridiag-}\mathcal{N}}$, then*

$$I(\xi; \eta) = -\frac{1}{2} \sum_j \log(1 - \rho_j^2) \tag{10}$$

*Proof of Corollary 4.12.* Under the proposed assumptions, $\log \det \Sigma_{\xi, \xi} = \log \det \Sigma_{\eta, \eta} = 0$, so $I(\xi; \eta) = -\frac{1}{2} \log \det \Sigma$. The matrix $\Sigma$ is block-diagonalizable via the following permutation:

$$(\xi_1, \ldots, \xi_{d_\xi}, \eta_1, \ldots, \eta_{d_\eta}) \mapsto (\xi_1, \eta_1, \xi_2, \eta_2, \ldots),$$

with the blocks being

$$\Sigma_{\xi_j, \eta_j} = \begin{bmatrix} 1 & \rho_j \\ \rho_j & 1 \end{bmatrix}$$

The determinant of block-diagonal matrix is a product of the block determinants, so $I(\xi; \eta) = -\frac{1}{2} \sum_j \log(1 - \rho_j^2)$. $\square$

**Statement 4.13** (Canonical correlation analysis)**.** *Let $(\xi, \eta) \sim \mathcal{N}(m, \Sigma)$, where $\Sigma$ is non-singular. There exist invertible affine mappings $\varphi_\xi$, $\varphi_\eta$ such that $(\varphi_\xi(\xi), \varphi_\eta(\eta)) \sim \mu \in S_{\text{tridiag-}\mathcal{N}}$. Due to Theorem 2.1, the following also holds: $I(\xi; \eta) = I(\varphi_\xi(\xi); \varphi_\eta(\eta))$.*

*Proof of Statement 4.13.* Let us consider centered $\xi$ and $\eta$, as shifting is an invertible affine mapping. Note that $\Sigma_{\xi, \xi}$ and $\Sigma_{\eta, \eta}$ are positive definite and symmetric, so the following symmetric matrix square roots are defined: $A = \Sigma_{\xi, \xi}^{-1/2}$, $B = \Sigma_{\eta, \eta}^{-1/2}$. By applying these invertible linear transformations to $\xi$ and $\eta$ we get

$$\text{cov}(A\xi, B\eta) = \begin{bmatrix} \Sigma_{\xi, \xi}^{-1/2} \Sigma_{\xi, \xi} (\Sigma_{\xi, \xi}^{-1/2})^T & \Sigma_{\xi, \xi}^{-1/2} \Sigma_{\xi, \eta} (\Sigma_{\eta, \eta}^{-1/2})^T \\ \Sigma_{\eta, \eta}^{-1/2} \Sigma_{\eta, \xi} (\Sigma_{\xi, \xi}^{-1/2})^T & \Sigma_{\eta, \eta}^{-1/2} \Sigma_{\eta, \eta} (\Sigma_{\eta, \eta}^{-1/2})^T \end{bmatrix} = \begin{bmatrix} I & C \\ C^T & I \end{bmatrix},$$

where $C = \Sigma_{\xi, \xi}^{-1/2} \Sigma_{\xi, \eta} (\Sigma_{\eta, \eta}^{-1/2})^T$.

Then, the singular value decomposition is performed: $C = URV^T$, where $U$ and $V$ are orthogonal, $R = \text{diag}(\{\rho_j\})$. Finally, we apply $U^T$ to $A\xi$ and $V^T$ to $B\eta$:

$$\text{cov}(U^T A\xi, V^T B\eta) = \begin{bmatrix} U^T U & U^T CV \\ (U^T CV)^T & V^T V \end{bmatrix} = \begin{bmatrix} I & R \\ R^T & I \end{bmatrix},$$

Note that $U^T A$ and $V^T B$ are invertible. $\square$

**Statement 4.14.** *Let $(\xi, \eta) \sim \mathcal{N}(0, \Sigma) \in S_{\text{tridiag-}\mathcal{N}}$, $\{z_k\}_{k=1}^{N} \subseteq \mathbb{R}^{d_\xi + d_\eta}$. Then*

$$\mathcal{L}_{\mathcal{N}(0, \Sigma)}(\{z_k\}) = I(\xi; \eta) + \mathcal{L}_{\mathcal{N}(0, I)}(\{\Sigma^{-1/2} z_k\}),$$

*where (implying $\rho_j = 0$ for $j > \min\{d_\xi, d_\eta\}$)*

$$\Sigma^{-1/2} = \underbrace{\left[\begin{array}{ccc|ccc} \alpha_j + \beta_j & & & \alpha_j - \beta_j & & \\ & \ddots & & & \ddots & \\ \hline \alpha_j - \beta_j & & & \alpha_j + \beta_j & & \\ & \ddots & & & \ddots & \end{array}\right.}_{d_\xi} \underbrace{\phantom{\begin{array}{ccc}\\\\\\\\\end{array}}}_{d_\eta}$$

$$\alpha_j = \frac{1}{2\sqrt{1 + \rho_j}}$$

$$\beta_j = \frac{1}{2\sqrt{1 - \rho_j}}$$

*and $I(\xi; \eta)$ is calculated via* (10).

*Proof of Statement 4.14.* Note that $\Sigma$ is positive definite and symmetric, so the following symmetric matrix square root is defined: $\Sigma^{-1/2}$. This matrix is a normalization matrix: $\mathrm{cov}(\Sigma^{-1/2}(\xi, \eta)) = \Sigma^{-1/2} \Sigma (\Sigma^{-1/2})^T = I$.

According to the change of variable formula,

$$\mathcal{L}_{\mathcal{N}(0,\Sigma)}(\{z_k\}) = \log \det \Sigma^{-1/2} + \mathcal{L}_{\mathcal{N}(0,I)}(\{\Sigma^{-1/2} z_k\})$$

As $\Sigma_{\xi,\xi} = I_{d_\xi}$ and $\Sigma_{\eta,\eta} = I_{d_\eta}$, from the equation (8) we derive

$$I(\xi; \eta) = -\frac{1}{2} \log \det \Sigma = \log \det \Sigma^{-1/2}$$

Finally, it is sufficient to validate the proposed closed-form expression for $\Sigma^{1/2}$ in the case of $2 \times 2$ matrices, as $\Sigma$ is block-diagonalizable (with $2 \times 2$ blocks) via the following permutation:

$$M : (\xi_1, \ldots, \xi_{d_\xi}, \eta_1, \ldots, \eta_{d_\eta}) \mapsto (\xi_1, \eta_1, \xi_2, \eta_2, \ldots),$$

Note that

$$\begin{bmatrix} \alpha + \beta & \alpha - \beta \\ \alpha - \beta & \alpha + \beta \end{bmatrix}^2 = 2 \cdot \begin{bmatrix} \alpha^2 + \beta^2 & \alpha^2 - \beta^2 \\ \alpha^2 - \beta^2 & \alpha^2 + \beta^2 \end{bmatrix} = \frac{1}{1 - \rho^2} \begin{bmatrix} 1 & -\rho \\ -\rho & 1 \end{bmatrix}$$

$$\frac{1}{1 - \rho^2} \begin{bmatrix} 1 & -\rho \\ -\rho & 1 \end{bmatrix} \cdot \begin{bmatrix} 1 & \rho \\ \rho & 1 \end{bmatrix} = \begin{bmatrix} 1 & 0 \\ 0 & 1 \end{bmatrix}$$

$\square$

## B    Non-Gaussian-based tests

As our estimator is based on Gaussianization, it seems natural that we observe good performance in the experiments with synthetic data acquired from the correlated Gaussian vectors via invertible transformations. Possible bias towards such data can not be discriminated via the *independency* and *self-consistency* tests, and hard to discriminate via the *data-processing* test proposed in [14, 20] for the following reasons:

1. *Independency* test requires $\hat{I}(X; Y) \approx 0$ for independent $X$ and $Y$. In such case, as $\mathrm{cov}(f_X(X), f_Y(Y)) = 0$ for any measurable $f_X$ and $f_Y$, $\hat{I}_{\mathrm{MIENF}}(X; Y) \approx 0$ in any meaningful scenario (no overfitting, no ill-posed transformations), regardless of the marginal distributions of $X$ and $Y$.

2. *Self-consistency* test requires $\hat{I}(X; Y) \approx \hat{I}(g(X); Y)$ for $X, Y$ and $g$ satisfying Theorem 2.1. In our setup, this test only measures the ability of normalizing flows to invert $g$, and provides no information about the quality of $\hat{I}(X; Y)$ and $\hat{I}(g(X); Y)$.

   Moreover, as we leverage Algorithm 1 with the Gaussian base distribution for the dataset generation, we somewhat test our estimator for the self-consistency.

3. *Data-processing* test leverages the *data processing inequality* [35] via requiring $\hat{I}(X; Y) \geq \hat{I}(g(X); Y)$ for any $X, Y$ and measurable $g$. Theoretically, this test may highlight the bias of our estimator towards binormalizable data. However, this requires constructing $X, Y$ and $g$, so $X$ and $Y$ are not binormalizable, $g(X)$ and $Y$ are and $\hat{I}(X; Y) < \hat{I}(g(X); Y)$, which seems challenging to achieve.

That is why we use two additional, non-Gaussian-based families of distributions with known closed-form expressions for MI and easy sampling procedures: *multivariate Student distribution* [52] and *smoothed uniform distribution* [14].

In the following subsections, we provide additional information about the distributions, closed-form expressions for MI and sampling procedures.

## B.1 Multivariate Student distribution

Consider $(n+m)$-dimensional $(\tilde{X}; \tilde{Y}) \sim \mathcal{N}(0, \Sigma)$, where $\Sigma$ is selected to achieve $I(\tilde{X}; \tilde{Y}) = \varkappa > 0$. Firstly, a correction term is calculated in accordance to the following formula:

$$c(k, n, m) = f(k) + f(k + n + m) - f(k + n) - f(k + m), \quad f(x) = \log \Gamma \left( \frac{x}{2} \right) - \frac{x}{2} \psi \left( \frac{x}{2} \right),$$

where $k$ is the number of degrees of freedom, $\psi$ is the digamma function. Secondly, $X = \tilde{X}/\sqrt{k/U}$ and $Y = \tilde{Y}/\sqrt{k/U}$ are defined, where $U \sim \chi_k^2$. The resulting vectors are distributed according to the multivariate Student distribution with $k$ degrees of freedom. According to [52], $I(X; Y) = \varkappa + c(k, n, m)$. During the generation, $\varkappa$ is set to $I(X; Y) - c(k, n, m)$ to achieve the desired value of $I(X; Y)$.

Note that $I(X; Y) \neq 0$ even in the case of independent $\tilde{X}$ and $\tilde{Y}$, as some information between $X$ and $Y$ is shared via the magnitude.

## B.2 Smoothed uniform distribution

**Lemma B.1.** *Consider independent* $X \sim \mathrm{U}[0; 1]$, $Z \sim \mathrm{U}[-\varepsilon; \varepsilon]$ *and* $Y = X + Z$. *Then*

$$I(X; Y) = \begin{cases} \varepsilon - \log(2\varepsilon), & \varepsilon < 1/2 \\ (4\varepsilon)^{-1}, & \varepsilon \geq 1/2 \end{cases} \tag{14}$$

*Proof.* Probability density function of $Y$ (two cases):

$$(\varepsilon < 1/2): \qquad p_Y(y) = (p_X * p_Z)(y) = \begin{cases} 0, & y < -\varepsilon \vee y \geq 1 + \varepsilon \\ \frac{y+\varepsilon}{2\varepsilon}, & -\varepsilon \leq y < \varepsilon \\ 1, & \varepsilon \leq y < 1 - \varepsilon \\ \frac{1+\varepsilon-y}{2\varepsilon}, & 1 - \varepsilon \leq y < 1 + \varepsilon \end{cases}$$

$$(\varepsilon \geq 1/2): \qquad p_Y(y) = (p_X * p_Z)(y) = \begin{cases} 0, & y < -\varepsilon \vee y \geq 1 + \varepsilon \\ \frac{y+\varepsilon}{2\varepsilon}, & -\varepsilon \leq y < 1 - \varepsilon \\ \frac{1}{2\varepsilon}, & 1 - \varepsilon \leq y < \varepsilon \\ \frac{1+\varepsilon-y}{2\varepsilon}, & \varepsilon \leq y < 1 + \varepsilon \end{cases}$$

Differential entropy of a uniformly distributed random variable:

$$h(\mathrm{U}[a; b]) = \log(b - a)$$

Conditional differential entropy of $Y$ with respect to $X$:

$$h(Y \mid X) = \mathbb{E}_{x \sim X} h(Y \mid X = x) = \mathbb{E}_{x \sim X} h(Z + x \mid X = x)$$

As $X$ and $Z$ are independent,

$$\mathbb{E}_{x \sim X} h(Z + x \mid X = x) = \mathbb{E}_{x \sim X} h(Z + x) = \int\limits_0^1 \log(2\varepsilon) \, dx = \log(2\varepsilon) \tag{15}$$

Differential entropy of $Y$:

$$h(Y) = -\int\limits_{-\infty}^{\infty} p_Y(y) \, dy = \begin{cases} \varepsilon, & \varepsilon < 1/2 \\ (4\varepsilon)^{-1} + \log(2\varepsilon), & \varepsilon \geq 1/2 \end{cases} \tag{16}$$

The final result is acquired via substituting (15) and (16) into (1). $\qquad\square$

Equation (14) can be inverted:

$$\varepsilon = \begin{cases} (4 \cdot I(X;Y))^{-1}, & I(X;Y) < 1/2 \\ -W\left[-\frac{1}{2}\exp(-I(X;Y))\right], & I(X;Y) \geq 1/2 \end{cases},$$  (17)

where $W$ is the product logarithm function.

### B.3 Additional experiments

Recall that in Section 5, we do not evaluate other estimators on the tests discussed in this part of the Appendix. To address this, we provide additional results for MINE and DINE-Gaussian in Figure 5. We chose MINE as it is the best performing critic-based method judging by the results from Figure 3, and is widely considered as a decent modern baseline. We chose DINE-Gaussian as (a) this method also employs normalizing flows, and (b) we were not able to acquire reliable estimates via this method during the tests presented in Figure 3. The results are presented in Figure 5.

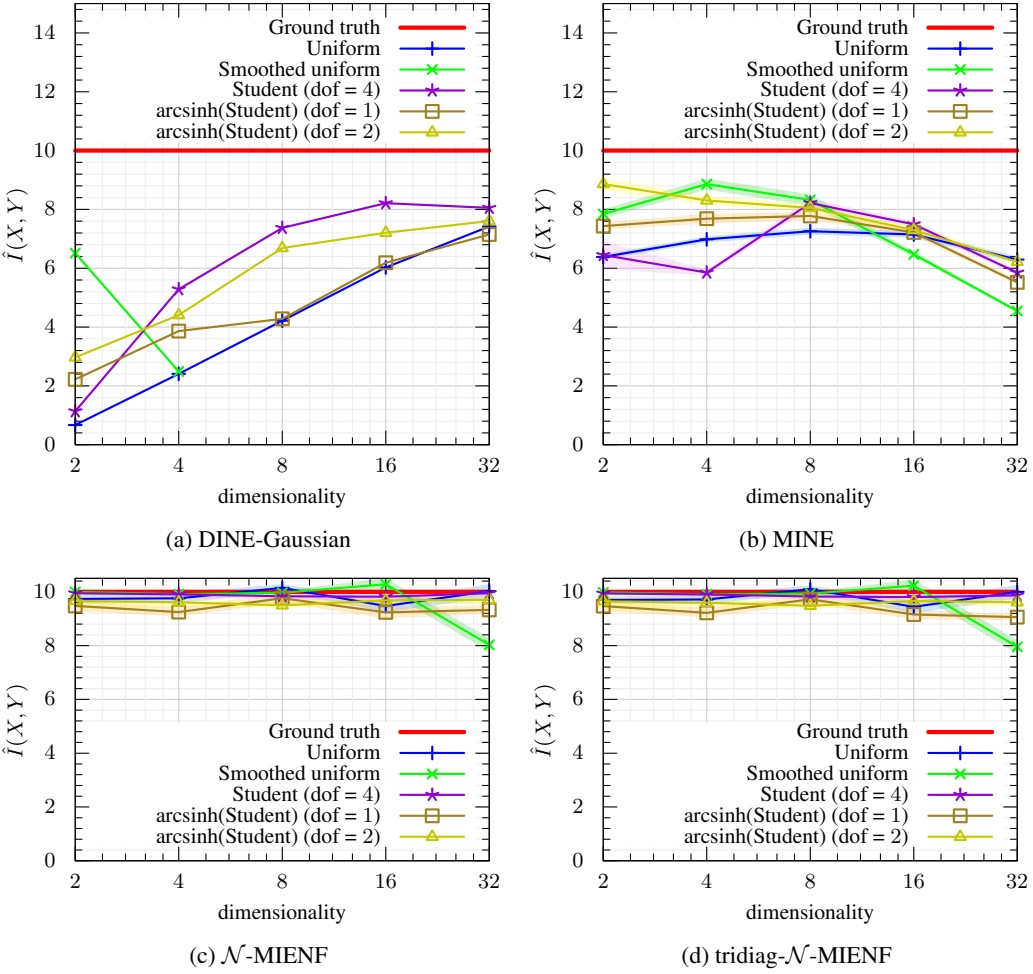

Figure 5: Additional tests with incompressible multidimensional data from Figure 4. $10 \cdot 10^3$ samples were used. We conduct only one experiment per point for DINE-Gaussian and MINE, acquiring CIs from epoch-wise averaging instead. Note that DINE-Gaussian failed to estimate MI for in the case of high-dimensional smoothed uniform distribution due to numerical instabilities. We also provide the plots for $\mathcal{N}$-MIENF and tridiag-$\mathcal{N}$-MIENF to facilitate the comparison.

## C  Overfitting and sample complexity

Due to the curse of dimensionality being a universal issue for MI-related tasks [13], our method, as any other NN-based estimator, is prone to overfitting in the case of small sample size. We illustrate this by performing an MI estimation on one of the benchmarks from Section 5, with the sampling being of normal and tiny size. We also conduct similar experiments with MINE. The resulting probability density and pointwise mutual information functions are presented in Figures 6 and 7.

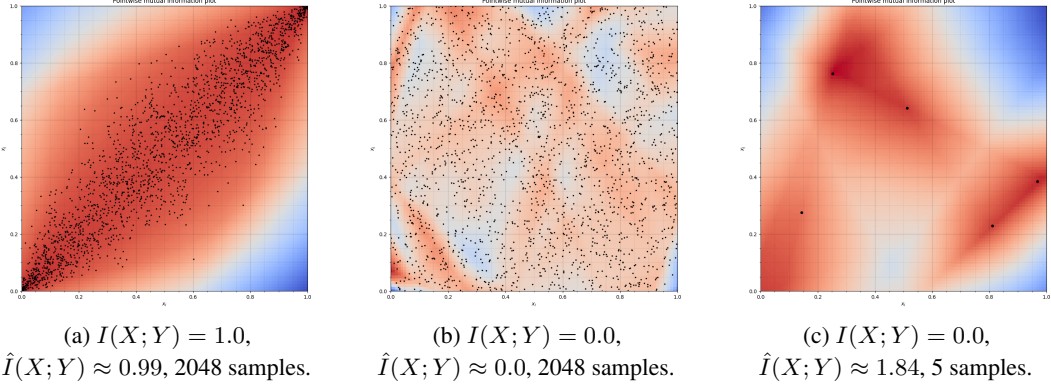

(a) $I(X;Y) = 1.0$, $\hat{I}(X;Y) \approx 0.99$, 2048 samples.

(b) $I(X;Y) = 0.0$, $\hat{I}(X;Y) \approx 0.0$, 2048 samples.

(c) $I(X;Y) = 0.0$, $\hat{I}(X;Y) \approx 1.84$, 5 samples.

Figure 6: Point-wise mutual information plots for MINE. Correlated uniform distribution is used, with varying ground truth MI and sampling size. Note that in the case of an insufficient sampling size, MINE "memorizes" the data points and "hallucinates" the relation between $X$ and $Y$, which severely increases the value of the MI estimate.

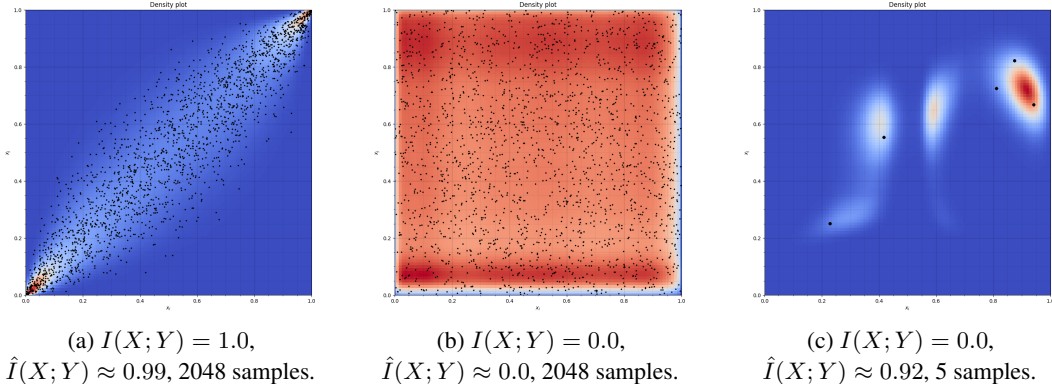

(a) $I(X;Y) = 1.0$, $\hat{I}(X;Y) \approx 0.99$, 2048 samples.

(b) $I(X;Y) = 0.0$, $\hat{I}(X;Y) \approx 0.0$, 2048 samples.

(c) $I(X;Y) = 0.0$, $\hat{I}(X;Y) \approx 0.92$, 5 samples.

Figure 7: Probability density function plots for tridiag-$\mathcal{N}$-MIENF. Correlated uniform distribution is used, with varying ground truth MI and sampling size. Note that in the case of an insufficient sampling size, MIENF "memorizes" the data points and "hallucinates" the relation between $X$ and $Y$, which severely increases the value of the MI estimate.

## D  Information-theoretic disentanglement

To explore additional applications of our method, we consider the task of representation disentanglement, i.e., the process of separating data into independent variables with distinct semantic meaning. For this example, we use the MNIST dataset of handwritten digits [53].

Let $X$ be a random image of a handwritten digit. Consider a Markov kernel $X \rightarrow (X', X'')$ corresponding to a pair of random augmentations applied to $X$ (we use random translation, rotation, zoom, and shear from `torchvision.transforms`). Now consider the task of estimating $I(X'; X'')$ (MI between the two augmented versions of the same image). Note that tridiag-$\mathcal{N}$-MIENF estimates the MI and performs a nonlinear canonical correlation analysis simultaneously (because of the tridiagonal covariance matrix in the latent space). Moreover, $\{\rho_j\}$ (from Definition 4.11) represent

the dependence between the nonlinear components. Higher values of $\rho_j$ (and, as a consequence, of the per-component MI) are expected to correspond to the nonlinear components, which are invariant to the selected augmentations (e.g., width/height ratio of a digit, thickness of strokes, etc.). We also expect small values of $\rho_j$ to represent the components, which parametrize the augmentations used in our setup (e.g, translation, zoom, etc.).

To perform the experiment, we train tridiag-$\mathcal{N}$-MIENF on samples from $(X', X'')$. We then randomly select several images from $X$, acquire their latent representations, apply a small perturbation along the axes corresponding to the highest and the lowest values of (one axis at a time), and perform an inverse transformation to visualize the result. We observe the expected behavior. The results are provided in Figure 8. We use a convolutional autoencoder beforehand to reduce the dimensionality (the size of the bottleneck is $64$) and speed up the experiment.

# E    Technical details

In this section, we describe the technical details of our experimental setup: architecture of the neural networks, hyperparameters, etc.

For the tests described in Section 5, we use architectures listed in Table 2. For the flow models, we use the `normflows` package [54]. The autoencoders are trained via Adam [55] optimizer on $5 \cdot 10^3$ images with a batch size $5 \cdot 10^3$, a learning rate $10^{-3}$ and MAE loss for $2 \cdot 10^3$ epochs. The MINE/NWJ/Nishiyama critic network is trained via the Adam optimizer on $5 \cdot 10^3$ pairs of images with a batch size $512$, a learning rate $10^{-3}$ for $5 \cdot 10^3$ epochs. The GLOW normalizing flow is trained via the Adam optimizer on $10 \cdot 10^3$ images with a batch size $1024$, a learning rate decaying from $5 \cdot 10^{-4}$ to $1 \cdot 10^{-5}$ for $2 \cdot 10^3$ epochs. Nvidia Titan RTX was used to train the models. In any setup, each experiment took no longer than one hour to be completed. In the following repositories, we provide PyTorch implementations of the NN-based estimators we used: `https://github.com/VanessB/pytorch-kld` and `https://github.com/VanessB/pytorch-mienf`.

Table 2: The NN architectures used to conduct the tests in Section 5.

| NN | | Architecture |
|---|---|---|
| AEs, $16 \times 16$ ($32 \times 32$) images | $\times 1$: | Conv2d(1, 4, ks=3), BatchNorm2d, LeakyReLU(0.2), MaxPool2d(2) |
| | $\times 1$: | Conv2d(4, 8, ks=3), BatchNorm2d, LeakyReLU(0.2), MaxPool2d(2) |
| | $\times 2(3)$: | Conv2d(8, 8, ks=3), BatchNorm2d, LeakyReLU(0.2), MaxPool2d(2) |
| | $\times 1$: | Dense(8, dim), Tanh, Dense(dim, 8), LeakyReLU(0.2) |
| | $\times 2(3)$: | Upsample(2), Conv2d(8, 8, ks=3), BatchNorm2d, LeakyReLU(0.2) |
| | $\times 1$: | Upsample(2), Conv2d(8, 4, ks=3), BatchNorm2d, LeakyReLU(0.2) |
| | $\times 1$: | Conv2d(4, 1, ks=3), BatchNorm2d, LeakyReLU(0.2) |
| MINE, critic NN, $16 \times 16$ ($32 \times 32$) images | $\times 1$: | [Conv2d(1, 16, ks=3), MaxPool2d(2), LeakyReLU(0.01)]$^{\times 2 \text{ in parallel}}$ |
| | $\times 1(2)$: | [Conv2d(16, 16, ks=3), MaxPool2d(2), LeakyReLU(0.01)]$^{\times 2 \text{ in parallel}}$ |
| | $\times 1$: | Dense(256, 128), LeakyReLU(0.01) |
| | $\times 1$: | Dense(128, 128), LeakyReLU(0.01) |
| | $\times 1$: | Dense(128, 1) |
| GLOW, $16 \times 16$ ($32 \times 32$) images | $\times 1$: | 4 (5) splits, 2 GLOW blocks between splits, 16 hidden channels in each block, leaky constant $= 0.01$ |
| | $\times 1$: | Orthogonal linear layer |
| | $\times 4$: | RealNVP(AffineCouplingBlock(MLP($d/2$, 32, $d$)), Permute-swap) |
| RealNVP, $d$-dimensional data | $\times 6$: | RealNVP(AffineCouplingBlock(MLP($d/2$, 64, $d$)), Permute-swap) |

Here we do not explicitly define $g_\xi$ and $g_\eta$ used in the tests with synthetic data, as these functions smoothly map low-dimensional vectors to high-dimensional images and, thus, are very complex. A Python implementation of the functions in question is available in the supplementary code repository, see `https://github.com/VanessB/mutinfo`.

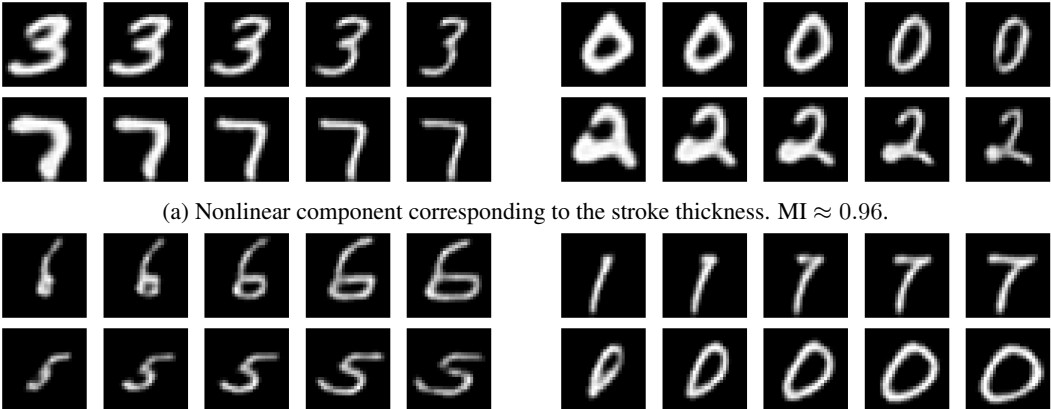

(a) Nonlinear component corresponding to the stroke thickness. MI ≈ 0.96.

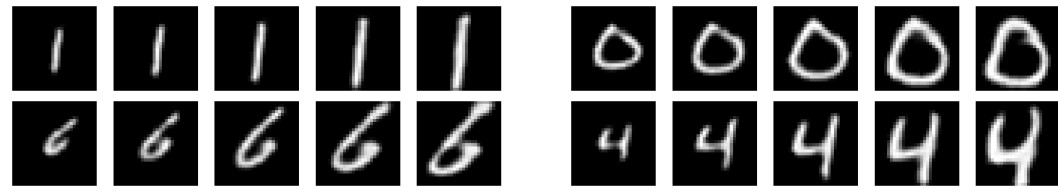

(b) Nonlinear component corresponding to the width of a digit. MI ≈ 0.75.

⋮

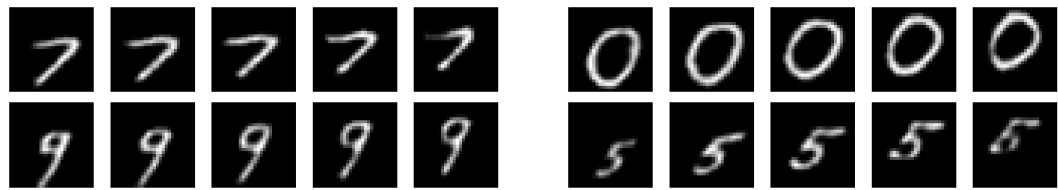

(c) Nonlinear component corresponding to zoom transformation. MI ≈ 0.002.

(d) Nonlinear component corresponding to vertical translation. MI < 0.001.

Figure 8: Results of an information-based nonlinear canonical correlation analysis performed on the MNIST handwritten digits dataset. The task of MI estimation between augmented (translated/rotated/...) versions of pictures is considered. Our method (the tridiagonal version) allows for simultaneous MI estimation and nonlinear independent components learning. We illustrate the semantics of the learned nonlinear components via small perturbations along the corresponding directions in the latent space. The center of each row contains an original, unperturbed picture; pictures to the left and to the right are the results of the perturbations. We also provide the values of per-component MI. Components with high MI represent the features, which are invariant to the selected augmentations.

