# OpenReview forum: "Mutual Information Estimation via Normalizing Flows"
_NeurIPS.cc/2024/Conference — NeurIPS 2024 poster_

### Official Review · Reviewer_2t4W · 2024-06-18

**Soundness:** 3
**Presentation:** 2
**Contribution:** 3
**Rating:** 4
**Confidence:** 4

**Summary:**

This paper proposes a family of mutual information (MI) estimators based on normalizing flows.  The general method proposes a Monte Carlo estimator that holds for any base distribution.  The authors refine this estimator for Gaussian base distributions, where the MI estimate can be calculated analytically.  Finally, the authors propose a simplified Gaussian base estimator with fixed mean and tridiagonal covariance matrices, which are simpler to compute compared to the general Gaussian case.

**Strengths:**

The paper addresses an important problem of estimating information-theoretic measures for arbitrarily complex target distributions.  Results in the paper appear technically sound despite a relatively high technical nature of the content.

**Weaknesses:**

The presentation of results is somewhat dense in places and the paper would benefit from more clarifying statements that contextualize the impact of results.  For example, Sec. 4.2 is rather dense with mathematical statements, but would benefit from remarks that inform the reader in plain English what impact these statements have.

Experimental validation is somewhat lacking in several aspects.  Firstly, the results in Fig. 1 are provided only with comparison to ground truth.  The authors claim that no baseline methods are provided for comparison as similar experiments have been conducted.  But without baseline comparison it is impossible to draw conclusions about efficacy of the proposed approach.  Secondly, the results are presented with minimial-to-no insight / discussion beyond simple validation metrics.  The authors should consider showing plots of the density estimates for several cases (i.e. at least the lower dimensional settings).  Finally, the authors do not compare to any existing flow-based methods such as DINE-Gaussian, which would be a useful comparison.

A simple oversight in this paper is the lack of any related work section.  Without this discussion it is difficult for the reader to contextualize the contributions of this paper relative to prior work.  This reviewer acknowledges that references are provided in the introduction, but the authors do not make clear how their current work advances the state-of-the-art.

**Detailed Comments**
*  I think the absolute values on the LHS of the equation in Corollary 3.2 are not necessary.  Similarly I think they are not needed in Corollary 4.4.
* Eq. (6) : rogue comma in denominator of first term
* Thm. 3.1 : change "defined on the same space as" to "absolutely continuous w.r.t."
* The justification of MLE is a little unclear, I think this is a result of corollary 3.2?
* Eq. (8) : This bound isn't directly proven as the difference in KL terms isn't shown to be nonnegative.  I think one can arrive at this conclusion for the same reason that the absolute values are not needed in Corollary 4.4?
* Sec. 5 : Consider showing example images of geometric shapes (i.e. in appendix) for the reader to better understand this experiment-in particular it is unclear what the authors mean by "incompressible" images

A low-priority high-level comment : The paper begins by arguing that the work is motivated by information-theoretic analysis of DNNs.  This argument is also revised in the discussion.  But  connection of this work to information-theoretic analysis of DNNs is never demonstrated.  The authors should consider motivating the work by more concrete applications such as representation learning or Bayesian experimental design, both of which employ MI estimates widely.

**Questions:**

See comments in Weaknesses section.

**Limitations:**

The authors outline limitations of the proposed methodology.

---

> ### Author Rebuttal · Authors · 2024-08-06
>
> We thank Reviewer **2t4W** for the work!
> We are glad to receive helpful criticism of our article.
> We further provide answers to the main points raised in the review.
>
> **Weaknesses:**
>
> 1. It is true that some parts of our work are dense with theoretical results.
>    As our manuscript has reached the page limit,
>    we hope to utilize the additional content page in the final version of the article to add more complementary explanation and discussion.
>
> 2. We agree that our article would benefit from the proposed amendments.
>    Some examples of density plots will be added to the article;
>    they are also available in the supplementary PDF (Figure 5); please, see the global reply.
>    An additional comparison with MINE and DINE-Gaussian has been also conducted;
>    please, refer to the same PDF, Figure 6.
>    In our setup, DINE-Gaussian did not yield reasonable results in the low-dimensional case,
>    severely underestimating the MI.
>
> 3. We believe that the overview provided in the introduction is mostly sufficient to familiarize the reader with the related work.
>    Nevertheless, we will additionally elaborate on this topic to explain the relation of our method to the previous estimators in more details.
>
> **Detailed comments:**
>
> 1. Unfortunately, the absolute values are required for the following reason.
>    In general, $D_\text{KL}(p_{\xi,\eta}||q_{\xi,\eta})$ and $D_\text{KL}(p_\xi \otimes p_\eta||q_\xi \otimes q_\eta)$ are incomparable.
>    For example, if $\xi=\eta$ under $p$ and $q$, then $D_\text{KL}(p_{\xi,\eta}||q_{\xi,\eta}) = D_\text{KL}(p_\xi||q_\xi) < 2 \cdot D_\text{KL}(p_\xi||q_\xi) = D_\text{KL}(p_\xi \otimes \eta||q_\xi \otimes q_\eta)$.
>    Conversely, if $p_\xi \otimes p_\eta = q_\xi \otimes q_\eta$ but $p_{\xi,\eta} \neq q_{\xi,\eta}$,
>    we have $D_\text{KL}(p_{\xi,\eta}||q_{\xi,\eta}) > 0 = D_\text{KL}(p_\xi \otimes \eta||q_\xi \otimes q_\eta)$.
>    Thus, the proposed bound is tight in both ways.
> 2. Thank you, the typo will be corrected.
> 3. Thank you again, the text will be changed.
> 4. Likelihood maximization is equivalent to $D_\text{KL}$ minimization (see [31]),
>    which tightens the bound in Corollary 3.2 and Corollary 4.3.
>    We will add this clarification to the manuscript.
> 5. To derive the bounds in question, one also have to utilize the monotonicity property of the KLD
>    ($D_\text{KL}(p_{X,Y}||q_{X,Y}) \geq D_\text{KL}(p_X||q_X)$).
>    This is reflected in our proofs of the bounds.
>    Please, refer to lines 780-781 (Appendix, proof of Corollary 3.2).
> 6. Thank your for the suggestion! The pictures will be added to the article.
>    We will also add them to the supplementary PDF of the general reply.
>
>    Please, also note that we do not call synthetic images "incompressible";
>    this term is used to describe the datasets for experiments in Figure 1.
>    These datasets have no low-dimensional latent structure.
> 7. It is true that adding more examples of applications will make the motivation of our work more clear.
>    This will be done in the next revision.
>
> We again sincerely thank Reviewer **2t4W** for carefully reading our article and providing us with useful comments and helpful criticism.
> We will be glad to address any further concerns if they arise.

---

> ### Author Response · Authors · 2024-08-12
> **Awaiting your reply**
>
> Dear Reviewer **2t4W**,
>
> Once again, thank you very much for your detailed review and the time you spent. As the end of the discussion period is approaching, we would like to ask if the concerns you raised have been addressed. We hope that you find our responses useful and would love to engage with you further if there are any remaining questions.
>
> We understand that the discussion period is short, and we sincerely appreciate your time and help!

---

### Official Review · Reviewer_m9Tu · 2024-07-03

**Soundness:** 3
**Presentation:** 4
**Contribution:** 4
**Rating:** 7
**Confidence:** 4

**Summary:**

This work presents an elegant and sound methodology for the estimation of mutual information (MI) in the context of high-dimensional continuous random variables (vectors).

The key intuition for the proposed methodology is that MI is an invariant measure under smooth injective mappings. Then, the authors define such mappings as trainable normalizing flows, which are used to transform the original data distribution in such a way that computing MI on such learned transformation is easy.

First, the authors propose a general method, which requires training normalizing flows for each random vector (say, $f_X$ and $f_Y$), and a simple model $q$ to approximate the PDF in latent space which has tractable point-wise MI. This approach allows to build a MI estimator whereby models $q$, $f_X$ and $f_Y$ are trained according to maximum likelihood.
For the general method, the authors show that the estimator is consistent, as well as they derive bounds that depend on the KL divergence between the the PDF of the transformed input vectors and the PDF of the mode $q$.

Then, the authors refine their general method with the goal of practicality. In this case, they restrict model $q$ to belong the family of multivariate Gaussian distributions. This allows to estimate MI via closed form-expressions, and to derive the optimal model $q$ in closed form as well. Furthermore, the authors show how to derive better non-asymptotic bounds and the variance of the proposed MI estimator.
In practice, this method requires training the cartesian product of $f_X \times f_Y$ as a single normalizing flow, and maximize the log likelihood of the joint sampling of input data, using as base distributions the whole set of Gaussian distributions.
It is important to notice that the authors are well aware of some issues that might affect the proposed Gaussian base distribution method. As such, they refine their method by considering a special kind of base distributions whereby the co-variances of the Gaussian base are sparse, tridiagonal and block-diagonalizable. Overall, this boils down to simplifying model $q$ from the general method described above by requiring the estimate of parameters $\rho$ that are reminiscent of non-linear canonical correlation analysis.

Experiments on synthetic data complement the methodological part of the paper, with the goal of assessing the quality of the proposed estimator in difficult setups, that is, with input data distributions that are high-dimensional, exhibiting long tailed distributions, and for which MI can be high.
Experiments are built from prior work that defined a set of benchmarks to compare available MI estimators.
Results indicate that, when ground truth is known, the proposed technique is stable, scales to high-dimensions and can cope with difficult data distributions.

**Strengths:**

* I really commend the very clear and sound mathematical treatment of both the general and the refined method. It is easy to follow, definitions are clear, theorems are well built and well commented. Also, consistency and bounds for the various proposed estimator variants are proposed, which is not always the case for competing methods

* This work is well rooted in the literature, and propose a different approach (which holds some similarity with recent work such as [22]) for MI estimation, which is sound and practical

* The authors did a good job in spelling out the limitations of their method, and came up with refinements that overcome such problems

**Weaknesses:**

* The computational scalability of the proposed method is not discussed in detail. The authors focus on requiring a simple model for $q$, in terms of parameter count, but do not spend the same energy in discussing the complexity of learning the single normalizing flow as a surrogate of the cartesian product of individual (in terms of input variables) flows. In the general case, the proposed method requires three models to be learned: two normalizing flows and the model $q$. This is in contrast to the recent literature such as the work in [22] cited by the authors, whereby amortization techniques are used to learn a unique model.

* It would be interesting to discuss in more detail the role of model $q \in \mathcal{Q}$. What is written in 106 to 110 is an excellent starting point, which culminates in Theorem 3.1 and Corollary 3.2, as well as Theorem 3.3 on the consistency of the estimator. What I am wondering about is the relationship between the “quality of $q$” and the sample complexity of the estimator. It is well known that, in general, neural estimation of MI suffers from high sample complexity, and since the proposed method falls in the family of neural estimators, a detailed discussion (or maybe experiments) on sample complexity seems missing

* The experiments presented in this paper are compelling and aligned with the literature. Nevertheless, since the narrative of this work is centered around practicality of the proposed method, it would have been a nice addition to have at least one experiment on realistic data.

**Questions:**

* Based on the comment about scalability above, would it be possible to provide a detailed discussion about the computational complexity of the proposed method (both in the general and in the refined variants)?

* Based on the comment on sample complexity above, can you help me better understand the implications of the quality of $q$ (and as a matter of fact, also of the normalizing flows) for cases in which only a limited amount of samples from the input distributions $X$ and $Y$ are available?

* Which problems, if any, do you foresee in the application of the proposed method on realistic data?

* In some application scenarios, MI has been used in conjunction with optimization objectives to learn the parameters of a model, e.g., in representation learning [1], or in text-to-image alignment problems [2]. In such cases, point-wise mutual information is generally preferred. In your work, you show through expression (5), that the same methodology can be applied to point-wise mutual information estimation. Can you elaborate more on this aspect?

[1] Mohamed Ishmael Belghazi, Aristide Baratin, Sai Rajeshwar, Sherjil Ozair, Yoshua Bengio, Aaron Courville, and Devon Hjelm. Mutual information neural estimation. In Proceedings of the 35th International Conference on Machine Learning, 2018.

[2] Xianghao Kong, Ollie Liu, Han Li, Dani Yogatama, and Greg Ver Steeg. Interpretable diffusion via information decomposition. In The Twelfth International Conference on Learning Representations, 2024.

**Limitations:**

Yes, these have been thoroughly discussed throughout the paper and in a dedicated section.

---

> ### Author Rebuttal · Authors · 2024-08-06
>
> Dear Reviewer **m9Tu**, thank you sincerely for your profound and comprehensive review!
> In the following text, we provide responses to your questions.
> We hope that all your concerns are properly addressed.
>
> **Weaknesses:** for 1 and 2 please refer to the questions section.
>
> 3. In order to assess the quality of our method and compare it to other approaches,
>    synthetic datasets had to be utilized, as they are the only option with available ground truth MI.
>    To improve the tests, we tried to reproduce the main challenges of realistic data:
>    high dimensionality and latent structure.
>    It is, however, true that our work would still benefit from experiments on realistic data.
>
>    To partially address this issue, we employ our method to conduct a nonlinear canonical correlation analysis of the handwritten digits dataset.
>    We describe the experiment and provide the results in the general reply.
>    Please, note that this experiment was conducted under the time constraints of the rebuttal process.
>
> **Questions:**
>
> 1. We believe that having three models instead of one should not be a significant problem if such approach allows for less learnable parameters to be used.
>    In contrast to a conventional approach, where three normalizing flows are used to estimate $h(X)$, $h(Y)$ and $h(X,Y)$ separately,
>    our refined method should be as expensive as learning only two flows for modeling $p_X$ and $p_Y$
>    (as we employ a Cartesian product of two flows).
>
>    The general method additionally requires learning (possibly complex) $q$,
>    but this approach still should not be as expensive as training a whole separate flow to model the joint distribution $p_{X,Y}$ from scratch.
>    That is because the joint distribution should be already partially disentangled and simplified by $f_X \times f_Y$.
>
>    In [22], several clever tricks are employed to learn only one model.
>    More specifically, manipulating the conditioning parameter $c$ in MINDE-C or diffusion speed modulators $\alpha,\beta$ in MINDE-J
>    allows for using a unique score network to model all the required distributions.
>    One is able to apply similar tricks to normalizing flows by using autoregressive flows [a,b],
>    but such approach is incompatible with our idea, as it can not be implemented via a Cartesian product of two flows.
>    However, this amortization technique can be applied to $q$.
>
>    We also note that all three models are learned in a single training loop with a shared loss function.
>    Thus, this is basically a single model consisting of three blocks.
>
>    Finally, the complexity of $q$ can be gradually increased (e.g., more components are added to a Gaussian mixture) to ensure that the most simple model is used.
>    This procedure should be fairly cheap, as it does **not** require retraining $f_X$ and $f_Y$ every time the complexity of $q$ is increased.
>    This can be interpreted as a gradual unfreezing of some parts of the model.
>
>    *Summary:* we believe that our method is as expensive as modeling just $p_{X,Y}$ in the worst case scenario.
>    The same should be true for both methods from [22].
>
> 2. In this work, we decided to focus on the refined approach for several reasons:
>    mathematical elegance, better error bounds and relatively decent performance.
>    Nevertheless, we also consider the question of sample complexity of the general method to be very important and interesting.
>    Although we admit that a proper investigation of this topic requires conducting an additional set of experiments,
>    we still would like to provide some related discussion.
>
>    It is indeed known that any consistent MI estimator requires an exponentially (in dimension) large number of samples [b,c].
>    This fact is related to the high sample complexity of the PDF estimation task.
>    One may expect $\mathcal{N}$-MIENF to be less prone to the curse of dimensionality,
>    as it is a fairly restricted model.
>    This, of course, comes at a cost of less expressive power.
>    On the other hand, increasing the complexity of $q$ in the general method makes the model less regularized and more expressive,
>    which increases the effects of the curse of dimensionality, but also improves the quality of the estimate for large datasets.
>    Thus, one can expect the sample complexity to increase as a consequence of using more expressive $q$.
>
>    Finally, the sample complexity also depends on $f_X$ and $f_Y$.
>    More expressive flows may require large numbers of samples to avoid overfitting.
>    In the case of small, but high-dimensional datasets NN-based MI estimators tend to "memorize" pairs of samples from $p_{X,Y}$,
>    yielding extremely high MI estimates.
>    We demonstrate this behaviour in the supplementary PDF, see the global reply.
>
>    *Summary:* using less expressive $q$, $f_X$ and $f_Y$ may serve as a regularization,
>    thus partially decreasing the sample complexity at a cost of lower accuracy for large datasets.
>
> 3. Due to the restrictive nature of normalizing flows, they are relatively expensive and less stable to train.
>    They are also inferior to diffusion models in terms of generative capabilities.
>    We believe that all these nuances may make it harder to apply our method to real data of very high dimensionality and complex structure.
>    We consider combining our binormalization method with diffusion models to alleviate these difficulties.
>
> 4. Yes, our method is perfectly suitable for PMI estimation.
>    Moreover, it is even possible to calculate PMI via a closed-form expression
>    while using less restricted $q$ (e.g. Gaussian mixture).
>    This allows us to achieve both consistency and simplicity of the PMI estimator.
>
> We hope that we were able to answer the raised concerns.
> These answers will be incorporated into the manuscript.
> We again sincerely thank Reviewer **m9Tu** for carefully reading our article and providing us with useful comments,
> helpful criticism and interesting questions!
> We will be glad to answer any further questions if they arise.

---

> > ### Comment · Reviewer_m9Tu · 2024-08-11
> > **Thank you for the rebuttal**
> >
> > Dear authors,
> > I've read your rebuttal and I am satisfied with the comments, additional details, and additional experiments.
> > I will keep my accept score, as I think this is very solid work and a nice contribution to the research community.

---

> > > ### Author Response · Authors · 2024-08-11
> > > **Thank you**
> > >
> > > Dear Reviewer **m9Tu**, we are glad that we were able to address your concerns. We would like to thank you again for the work!

---

> ### Author Response · Authors · 2024-08-06
> **Additional references**
>
> [a] Kingma, D. P., Salimans, T., Jozefowicz, R., Chen, X., Sutskever, I., and Welling, M.
> Improved variational inference with inverse autoregressive flow.
> In Advances in neural information processing systems, pp. 4743–4751, 2016.
>
> [b] Atanov, Andrei, Alexandra Volokhova, Arsenii Ashukha, Ivan Sosnovik and Dmitry P. Vetrov.
> Semi-Conditional Normalizing Flows for Semi-Supervised Learning. ArXiv abs/1905.00505, 2019.
>
> [c] Z. Goldfeld, K. Greenewald, J. Niles-Weed, and Y. Polyanskiy.
> Convergence of smoothed empirical measures with applications to entropy estimation.
> IEEE Transactions on Information Theory, 66(7):4368–4391, 2020.
>
> [d] David McAllester and Karl Stratos. Formal limitations on the measurement of mutual information.
> Proceedings of the Twenty Third International Conference on Artificial Intelligence and Statistics, 2020

---

### Official Review · Reviewer_4miK · 2024-07-11

**Soundness:** 4
**Presentation:** 4
**Contribution:** 3
**Rating:** 7
**Confidence:** 4

**Summary:**

The authors aim to provide an automatic method for performing an information-theoretic analysis of DNNs
However, calculating mutual information (MI) and differential entropy of high-dimensional data are extremely hard to estimate.
The authors propose modeling a joint distribution of RVs with a Cartesian product of normalizing flows, which allows for a direct estimation of MI. Through some minor restrictions on the base distribution, the authors derive a very computationally feasible approach, while still demonstrating good performance in their experiments.

**Strengths:**

Overall, the paper is strong, showing a high level of rigor and precision without overcomplicating the subject. In the general Gaussian case, they show that a lower bound can be calculated in close-form — a notable advantage over prior work. Further, the authors place reasonable restrictions on the base distribution in order to avoid the more computationally intense calculations (e.g., inverting covariance matrix). The experimental results are promising and even show that tridiag N-MIENF yields nearly the same results as N-MIENF (Figure 1). Additionally, the results in Figure 2 show not only good performance of the estimator, but also evidence of the bound holding.

**Weaknesses:**

- Line 145 this is not clear, consider explaining or removing.

- Why aren't results for tridiag N-MIENF shown in Figure 2? The tridiag N-MIENF method is a strong contribution if the experiments can sufficiently support that the simplified bound is not too determinantal to the performance.

- The paper's initial motivation describes tools for explainable AI and an information-theoretic analysis of deep neural networks, however none of the experiments or results apply the method in this setting. Instead the experiments estimate MI from toy data that is algorithmically generated with known MI. The experiments support the method, but the paper lacks a compelling view of what cheap, accurate estimation of MI truly unlocks.

**Questions:**

Statement 4.8 requires that fx and fy be block diagonal, however Statement 4.8 applies Corollary 4.3 and 4.4 which defines on these RVs not to be block diagonal. Please clarify if this is correct, and if so does it limit the derived bound?

**Limitations:**

In paragraph 238 and 285 you briefly mention extensions to injective normalizing flows, specifically citing [9] which shows normalizing flows for manifold learning. This may work, but depending on the type of mapping, that flow may be optimizing a lower bound (and not exact an likelihood calculation), which may have negative impacts on your MI estimator. It is fine if this is outside the scope of your paper, but you may want to mention potential limitations or challenges if you suggest this approach.

---

> ### Author Rebuttal · Authors · 2024-08-06
>
> We would like to deeply thank Reviewer **4miK** for reading the article and providing us with a profound review!
> We further provide answers to the main points raised in the review.
>
> **Weaknesses:**
>
> 1. We wanted to stress out that the biggest possible gap between the lower bound and the true value of MI is indeed achievable.
>    Thus, the bound in Corollary 4.4 is tight.
>    The same example from Remark 4.5 also shows that it is insufficient to marginally Gaussianize a pair random vectors,
>    as the MI estimation error may become arbitrary high.
>    We will explain this remark more thoroughly in the next revision.
>
> 2. This appears to be a small mistake in the legend, as the results are provided for tridiag-$\mathcal{N}$-MIENF.
>    Choosing the tridiagonal version over the basic was indeed motivated by the near-identical performance (Figure 1),
>    as well as the mathematical equivalence of the methods (see Statement 4.13).
>    To avoid clutter, we plot only one method in Figure 2,
>    as the results of both methods on synthetic images are also very close to each other.
>    The label will be changed to represent the right method.
>
> 3. In order to assess the quality of our method and compare it to other approaches,
>    synthetic datasets had to be utilized, as they are the only option with available ground truth MI.
>    To improve the tests, we tried to reproduce the main challenges of realistic data:
>    high dimensionality and latent structure.
>
>    It is true that our work would benefit from experiments in the setting of explainable AI and an information-theoretic analysis of DNNs.
>    However, we decided to leave these experiments for the future, and focus on refining and evaluating the method.
>    We believe that the resulting estimator is still a notable contribution to the field.
>
>    Nevertheless, to partially address this issue, we employ our method to conduct a nonlinear canonical correlation analysis of the handwritten digits dataset.
>    We describe the experiment and provide the results in the general reply.
>    Please, note that this experiment was conducted under the time constraints of the rebuttal process.
>
> **Questions:**
>
> We do not require $f_X$ and $f_Y$ to be block-diagonal.
> Instead, we only utilize the fact that $f=f_X \times f_Y$ has a block-diagonal Jacobian,
> as it is a Cartesian product of two mappings.
> This is just a small simplification,
> which is not of great importance.
>
> **Limitations:**
>
> This is a valid point, which we will address in the next revision.
> The quick answer is that exact likelihood calculation is not required in our method.
> However, optimization of the exact likelihood (instead of a lower bound) should tighten the bound from Corollary 3.2 better,
> which in turn should yield better MI estimates.
> We also consider extending our idea to non-likelihood based methods as a part of the future work.
> This discussion will be added to the limitations section.
>
> We again sincerely thank Reviewer **4miK** for carefully reading our article and providing us with helpful comments.
> We will be glad to answer any further questions if they arise.

---

> > ### Comment · Reviewer_4miK · 2024-08-14
> >
> > Thank you for answering my questions, that has helped to address any concerns I have. I appreciate you addressing the limitation I listed as well, your response makes sense and by not requiring exact likelihood there may be additional applications for your technique. I will keep my score, this is very nice work and presented very nicely.

---

> > > ### Author Response · Authors · 2024-08-14
> > > **Thank you**
> > >
> > > Dear Reviewer **4miK**, we are glad that you find our response satisfactory. We would like to thank you again for your review!

---

### Official Review · Reviewer_cdSA · 2024-07-13

**Soundness:** 3
**Presentation:** 3
**Contribution:** 3
**Rating:** 7
**Confidence:** 4

**Summary:**

This paper presents a new approach for estimating mutual information (MI) using normalizing flows. The authors provide a series of theoretical results and demonstrate numerical examples.

**Strengths:**

1. The paper has provided comprehensive theoretical discussions for the proposed approach. The presentation is easy to follow.
2. The analyses seem to be sound and clear though I did not check the derivations line by line.
3. The authors conducted various experiments to test the algorithm.

**Weaknesses:**

1. Though the theoretical development is self-contained, the paper could potentially benefit from experiments on real datasets with MI-related tasks.
2. The presentation could be improved by adding visual details, such as a diagram of the general algorithm or examples of synthetic images.

**Questions:**

1. Sec 4.4, would the double log (double exponential) introduced here cause numerical issues?

[Typo] Line 427, Ref[44]

**Limitations:**

The authors adequately addressed the limitations.

---

> ### Author Rebuttal · Authors · 2024-08-06
>
> Dear Reviewer **cdSA**, thank you for reading and reviewing our article carefully!
> In the following text, we provide responses to your questions.
> We hope that all the concerns are properly addressed.
>
> **Weaknesses:**
>
> 1. We agree that experiments with real datasets will further improve our work and complement the current results acquired on synthetic data.
>    Unfortunately, such data does not allow for a direct assessment of the quality of the MI estimators.
>    That is why we focus on synthetic tests, while still trying to reproduce the main challenges of realistic data:
>    high dimensionality and latent structure.
>
>    To partially address this issue, we employ our method to conduct a nonlinear canonical correlation analysis of the handwritten digits dataset.
>    We describe the experiment and provide the results in the general reply.
>    Please, note that this experiment was conducted under the time constraints of the rebuttal process.
>
> 2. Two diagrams of both MI estimatiors will be added to the article, as well as some examples of the synthetic data.
>    We have also added these figures to the global reply PDF.
>
> **Questions:**
>
> 1. Note that we store $w_j$ and compute $\hat I = \sum_j e^w_j$ and $\rho_j = \sqrt{1 - \exp(- 2 \cdot e^{w_j})}$.
>    As $-2 \cdot e^{w_j} < 0$, $\exp(-2\cdot e^{w_j}) \in (0;1)$, which partially secures the method from numerical instabilities.
>    Some other issues might arise from the gradient calculation and backpropagation,
>    but we did not observe them.
>    Moreover, the training process for tridiag-$\mathcal{N}$-MIENF was relatively stable,
>    with the loss function plot being quite smooth and almost monotonic.
>
>    One can also use other smooth mappings from $\mathbb{R}$ to $(0;+\infty)$ to parametrize $\rho_j$.
>    For example, if the softplus function is employed, $\hat I = \sum_j \log(1 + e^{w_j})$ and $\rho_j = \sqrt{1 - 1/\sqrt{1 + e^{w_j}}}$,
>    which eliminates one of the exponential.
>    We will also reflect this approach in the final revision of the work.
>
> 2. The typo will be corrected, thank you!
>
> We hope that we were able to answer all the raised concerns. We again thank Reviewer **cdSA** for the work!
> If there are any questions left, we will be glad to answer them.

---

> > ### Comment · Reviewer_cdSA · 2024-08-12
> >
> > I have read through the rebuttal. Thanks for the detailed response and clarifications. The diagram and new experiment in the PDF are helpful for better readability. As a comment, I believe further discussions on the implementation/design choices can lead to some interesting future works, though the current form suffices the purpose of this manuscript.

---

> > > ### Author Response · Authors · 2024-08-12
> > > **Thank you**
> > >
> > > Dear Reviewer **cdSA**, we are glad that you are satisfied with our answer. We would like to thank you again for the review and for suggesting an interesting direction for the future work!

---

### Author Rebuttal · Authors · 2024-08-06

We would like to thank the reviewers for their time and effort to make our work better. To address the raised concerns, we answered each reviewer in individual messages below.

Some questions require supplementary materials to be submitted, including pictures, diagrams and additional experimental results. That is why we provide the following in the attached PDF:

1. Diagrams of the general and Gaussian-based methods. This will be added to the article.
2. Examples of the synthetic images used to assess the quality of different MI estimation methods. This will be added to the article.
3. Density plots (for MIENF) and pointwise mutual information plots (for MINE) to better understand how the methods work and how they perform in the case of tiny datasets. This will be added to the article, including additional plots and discussion.
4. Additional experiments with baseline MI estimators (MINE and DINE-Gaussian). The same datasets were used as in Figure 1 of our paper. This will be added to the article in an enhanced form, with confidence intervals, technical details and additional discussion.
5. An example experiment with real data: an information-based nonlinear canonical correlation analysis of the MNIST handwritten digits dataset. Please, note that this experiment was conducted to address some of the issues raised by the reviewers, under the time constraints of the rebuttal process. It will be reproduced in the next revision of the article with a thorough discussion of the experimental setup and results.

**Description of the experiment with real images:**

Let $ X $ be a random image of a handwritten digit. Consider a Markov chain $ X \to (Y, Z) $ corresponding to a pair of random augmentations applied to $ X $ (we use random translation, rotation, zoom and shear from `torchvision.transforms`). Now consider the task of estimating $ I(Y;Z) $ (MI between the two augmented versions of the same image). Note that tridiag-$\mathcal{N}$-MIENF estimates the MI and performs a nonlinear canonical correlation analysis simultaneously (because of the tridiagonal covariance matrix in the latent space). Moreover, $ \rho_j $ (from Definition 4.11) represent the dependence between the nonlinear components. Higher values of $ \rho_j $ (and, as a consequence, of the per-component MI) are expected to correspond to the nonlinear components, which are invariant to the selected augmentations (e.g., width/height ratio of a digit, thickness of strokes, etc.). We also expect small values of $ \rho_j $ to represent the components, which parametrize the used augmentations (e.g, translation, zoom, etc.).

To perform the experiment, we train tridiag-$\mathcal{N}$-MIENF on samples from $(Y,Z)$. We then randomly select several images from $X$, acquire their latent representations, apply a small perturbation along the axes corresponding to the highest and the lowest values of $\rho_j$ (one axis at a time), and perform an inverse transformation to visualize the result. We observe the expected behavior. The pictures are provided in the attached PDF, Figure 7.

We again would like to thank all the reviewers for the work. We hope that we were able to address most of the issues.

---

### Decision · Program_Chairs · 2024-09-25

**Decision:**

Accept (poster)

**Comment:**

This paper proposes a machine-learning-baesd method, termed mutual information estimation via normalizing flow (MIENF), to evaluate mutual information (MI) between random variables (rv's). The proposed method is to train a pair of normalizing flows, yielding a latent joint distribution whose marginals are Gaussians. As MI of such a pair of latent rv's is bounded from below by that of joint Gaussian rv's, the authors propose to use the lower bound as an estimate $\hat{I}_{𝒩-MIENF}$ of ML (equation (9)).
A variant $\hat{I}_{tridiag-𝒩-MIENF}$ (equation (11)) is also proposed on the basis of tridiagonalization of the covariance of the latents.

The rating/confidence scores of the reviewers were 7/4, 7/4, 7/4, and 4/4. Upon reading the paper myself, as well as the reviews and the author rebuttals, I also think that there are several weaknesses, some of which are summarized below. Most reviewers evaluated, however, that the significance of the contributions to the literature of neural-net-based estimation of MI would outweigh these weaknesses.
- The non-asymptotic error bounds (Corollaries 3.2 and 4.4) still depend on the true joint distribution $p_{\xi,\eta}$  (or its pushforward $\mathbb{P}\circ(f_X^{-1}\times f_Y^{-1})$ via $f_X\times f_Y$), so that one cannot evaluate them in practice without extensive Monte-Carlo integration. This paper does not provide any error bound that can be evaluated in a practical sense for the proposed estimator $\hat{I}:=\hat{I}_{𝒩-MIENF}$ itself.
Assuming that the normalizing flows $f_X,f_Y$ yield marginal Gaussianization, what one could say is that the proposed estimator $\hat{I}$ is a lower bound of the true MI (line 170), and one cannot tell anything about how large the estimation error $I-\hat{I}$ is.
- As **Reviewer m9Tu** commented, the sample complexity of the overall scheme, including what would be required for training of the normalizing flows $f_X,f_Y$, has not been discussed. I think that in a high-dimensional setting one would need a significant amount of samples for the training of normalizing flows, which would make the significance of the result (Lemma 4.9) of the asymptotic variance of the proposed estimator quite unclear, as it assumes $f_X,f_Y$ to be *fixed*.
- All the reviewers showed concern about the lack of experiments under more practical scenarios related with deep/representation learning.
- I would appreciate it if the authors would elaborate more on the consistency statement in Theorem 3.3. Here, one has to estimate $\hat{f}_X,\hat{f}_Y,\hat{q}$ in the non-parametric setting, and thus I think that consistency of maximum-likelihood estimation would not be so straightforward. The proof of Theorem 3.3 is so brief that it does not seem to be taking proper care of it. I would like to suggest to downgrade "Theorem" to "Claim" or somethink like this, unless a rigorous treatment of this issue is possible.
- In Definition 4.11, the crosscovariance matrix $\Sigma_{\xi,\eta}$ is assumed to be diagonal. What it exactly means should be clarified, because $\Sigma_{\xi,\eta}$ is not square when $d_\xi\not=d_\eta$.
Similarly, in Statement 4.14 one has to take care of the case $d_\xi\not=d_\eta$ in order to give a correct expression for $\Sigma^{-1/2}$.

Minor points:
- Line 132: This allows (a → u)s
- Line 138: $Z'$ is undefined.
- Definition 4.7: $S$ is undefined.
- Line 173: Let $(\xi,\eta)$ ha(s → ve)
- Lines 178, 223, 278; normalizing flow(s) setup
- Line 183: gradient descen(d → t) step
- Line 188: gradient ascen(d → t)
- Line 193: "ill-posed" might be "ill-conditioned".
- Line 198: I did not understand what "generalization ability" means. I guess that it should read "generality".
- Line 230: arbitrar(il)y close
- Table 1: What the RMSE means should be made clear. My guess is that it is the RMSE of the estimated MI relative to the true MI.
- Line 292: which can (can) be
- Line 790: arrive (to → at)
- Equation above line 808: $\hat{m}+b$ → ${\color{red}A}\hat{m}+b$
- Equation below line 811: $\Sigma_{\xi_i,\eta_i}$ → $\Sigma_{\xi_{\color{red}j},\eta_{\color{red}j}}$
- Line 813: $\Sigma_k$ → $\Sigma_j$
- Line 816: these invertible linear transformation(s)
- Equation (13): Put a comma after $(4\varepsilon)^{-1}$.
- Equation (16): The conditions are $I(X;Y){\color{red}>}1/2$ and $I(X;Y){\color{red}\le}1/2$.